# DECOMPOSED ATTENTION FUSION IN MLLMS FOR TRAINING-FREE VIDEO REASONING SEGMENTATION

**Su Ho Han**[1]* **Jeongseok Hyun**[1]* **Pilhyeon Lee**[2] **Minho Shim**[3]

**Dongyoon Wee**[3] **Seon Joo Kim**[1]

[1]Yonsei University  [2]Inha University  [3]NAVER Cloud

## ABSTRACT

Multimodal large language models (MLLMs) demonstrate strong video understanding by attending to visual tokens relevant to textual queries. To directly adapt this for localization in a training-free manner, we cast video reasoning segmentation as a video QA task and extract attention maps via rollout mechanism. However, raw attention maps are noisy and poorly aligned with object regions. We propose Decomposed Attention Fusion (DecAF), which refines these maps through two mechanisms: (1) contrastive object-background fusion and (2) complementary video-frame fusion. This method suppresses irrelevant activations and enhances object-focused cues, enabling direct conversion of attention maps into coarse segmentation masks. In addition, we introduce attention-guided SAM2 prompting for obtaining fine-grained masks. Unlike existing methods that jointly train MLLMs with SAM, our method operates entirely without retraining. DecAF outperforms training-free methods and achieves performance comparable to training-based methods on both referring and reasoning VOS benchmarks.

## 1 INTRODUCTION

In recent years, Multimodal Large Language Models (MLLMs) (Lin et al., 2023; Chen et al., 2024; Zhang et al., 2024; Bai et al., 2025; Wang et al., 2024a) have rapidly advanced, demonstrating strong performance on challenging video QA benchmarks (Mangalam et al., 2023; Fu et al., 2025). These advances reveal their ability to process temporal visual cues and perform complex reasoning over natural language queries. Such capabilities imply that MLLMs may also possess inherent localization ability in videos, enabling training-free video reasoning segmentation, a task that localizes objects corresponding to text-based queries requiring complex reasoning.

Recent studies (Yan et al., 2024; Bai et al., 2024; Gong et al., 2025b; Lin et al., 2025) have attempted to adapt MLLMs and segmentation foundation models (*e.g.*, SAM (Lin et al., 2024), SAM2 (Ravi et al., 2024)) for video reasoning segmentation by joint training through efficient fine-tuning methods such as LoRA (Hu et al., 2022). However, these methods require model-specific training and joint optimization of two foundation models, resulting in significant computation cost and limited generalization capability.

Meanwhile, a training-free approach, Loc-Head (Kang et al., 2025a), explores the localization ability of MLLMs in the image domain by selecting attention heads responsible for grounding. However, it assumes the presence of a single object referring object and selects heads based on spatial entropy, making extension to multi-object and temporal video data difficult. Moreover, it relies on heuristics to mitigate the visual attention sink phenomenon (Kang et al., 2025b), where certain regions consistently receive dominant attention scores regardless of the instruction, limiting its generalization across MLLMs. These observations motivate us to directly examine how attention mechanisms within MLLMs contribute to localization, without model-specific modification or training.

To obtain attention maps for object localization without relying on model- or task-specific design, we start with attention rollout (Abnar & Zuidema, 2020). Rollout aggregates attention weights across

---

*Equal contribution.
 Code is available at https://github.com/HYUNJS/DecAF

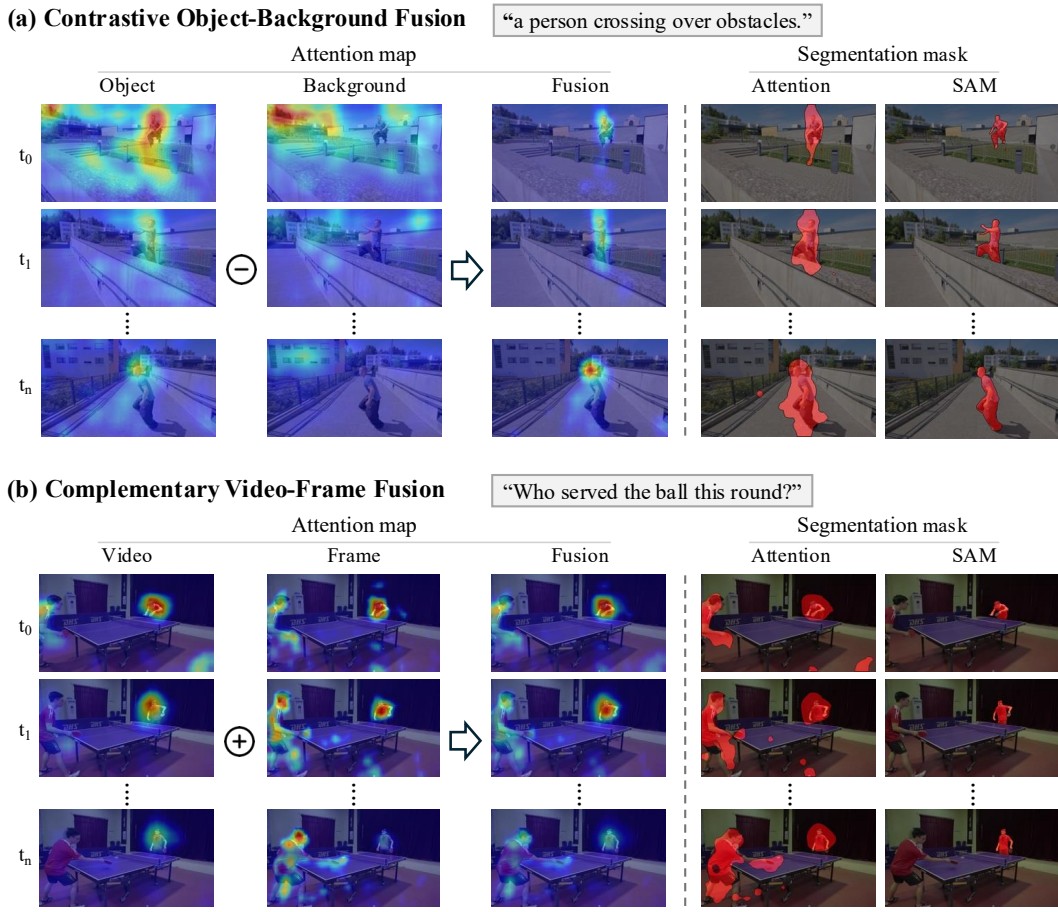

Figure 1: Visualization of our method. **(a)** Noise in irrelevant regions is suppressed by contrastive fusion with the background attention map. As shown in the first frame, background activations are removed, and the target object is emphasized. **(b)** Video attention map captures temporal cues, while frame attention map highlights object-centric details. Their fusion resolves conflicts (*e.g.*, identifying the server vs. the hitting player) and produces more consistent localization. The attention mask is obtained directly from the attention map while the SAM mask is generated by SAM2.

layers, revealing visual cues to which the MLLM attends when producing answers. Its applicability across attention-based MLLMs makes it a plausible approach to probing localization ability. However, since the rollout integrates signals from all heads, irrelevant regions and visual attention sinks often dominate, reducing the relative strength of object cues.

To overcome these limitations, we introduce *Decomposed Attention Fusion (DecAF)*, which suppresses noise and enhances object-focused signals by decomposing and fusing attention maps in two distinct ways. First, *contrastive object-background fusion* combines the object and background attention maps through a simple subtraction. The object attention map is obtained with a prompt focusing on the target object, while the background attention map is derived from a contrastive prompt excluding this object. This design effectively suppresses irrelevant activations and highlights the target object signal, as illustrated in Fig. 1 (a). Second, *complementary video-frame fusion* leverages the distinct strengths of video and frame attention in a multi-scale manner. Video attention captures temporal context, which is essential when the object is temporarily absent or requires temporal reasoning, but its coarse granularity limits performance on small objects. In contrast, frame attention provides object-centric, fine-grained cues but lacks temporal coherence. By combining these two attentions, this fusion maintains clearer object focus while also leveraging temporal context, resulting in more robust attention maps that accurately localize the target object across the video.

With the object localization attention map obtained from our two fusion methods, we first generate video object masks through simple thresholding, which provides reliable localization of the target object but remains coarse due to the low granularity of attention. To obtain denser masks, we extract point prompts from the attention map and apply SAM2 (Ravi et al., 2024). However, these coarse prompts, derived from spurious activations in the attention map, often produce false positives. To address this issue, we propose an attention consistency score that evaluates the alignment between the predicted mask and the underlying attention map, enabling unreliable segmentation masks to be filtered out. As shown in Fig. 1, this process transforms a noisy attention map into a precise and reliable segmentation mask.

We evaluate DecAF across three MLLM families and five datasets, including three referring VOS datasets (Khoreva et al., 2018; Seo et al., 2020; Ding et al., 2023) and two reasoning VOS datasets (Yan et al., 2024; Bai et al., 2024). DecAF consistently outperforms prior training-free approaches (Li et al., 2025; Kang et al., 2025a), both with and without SAM. In addition, the dense video object masks achieve performance comparable to training-based methods (Lai et al., 2024; Yan et al., 2024; Bai et al., 2024; Lin et al., 2025; Gong et al., 2025b;a). These results highlight that decomposed attention fusion offers a simple and effective framework for training-free video reasoning segmentation.

## 2 RELATED WORK

**Multimodal Large Language Models.** LLMs demonstrate powerful reasoning and cognition capabilities (Brown et al., 2020; Dubey et al., 2024; Yang et al., 2024), leading to the development of MLLMs (Wang et al., 2024b; Google, 2024; Team, 2024; Liu et al., 2024). These models, built on the transformer architecture (Vaswani et al., 2017), rely on the attention mechanism. Due to the quadratic cost of attention, some MLLMs firstly compress video tokens into a fixed number of tokens via a lightweight modules (Jin et al., 2024; Song et al., 2024; Maaz et al., 2024). However, this token compression inevitably sacrifices fine-grained spatial information, unlike LLaVA-style models (Liu et al., 2023), which use a linear projector to preserve dense spatial features. More recently, Qwen2VL (Wang et al., 2024b) further advances this line by supporting native-resolution video inputs, maintaining both aspect ratio and fine-grained visual details. In this work, we build on such models and focus on exploring the inherent localization ability of MLLMs.

**Text-conditioned Video Object Segmentation.** Early research on referring VOS (RVOS) focuses on localizing the target object from simple textual expressions, typically describing appearance. Datasets such as Ref-DAVIS (Khoreva et al., 2018) and Ref-YouTube-VOS (Seo et al., 2020) were designed for this setting and only cover single-object cases. More recently, MeViS (Ding et al., 2023) introduces motion-centric and more challenging scenarios, including cases where the referred object is absent or where multiple candidates match the expression.

With the advent of powerful MLLMs (Liu et al., 2023), video reasoning segmentation has emerged, targeting complex expressions that extend beyond appearance or motion cues and require reasoning over world knowledge and temporal context (Yan et al., 2024; Bai et al., 2024). To address this, existing approaches adapt pretrained MLLMs to RVOS via lightweight finetuning strategies such as LoRA (Hu et al., 2022), and integrate them with segmentation model such as SAM (Kirillov et al., 2023) for precise mask generation, often requiring full finetuning of the mask decoder (Gong et al., 2025b; Lin et al., 2025). In contrast, we leverage MLLMs and SAM in a training-free manner.

**Training-free Text-to-Visual Grounding with MLLMs.** Recently, MLLMs have been studied for training-free visual grounding tasks (Lin et al., 2024; Li et al., 2025; Kang et al., 2025a). VL-SAM (Lin et al., 2024) and TAM (Li et al., 2025) leverage the attention rollout mechanism (Abnar & Zuidema, 2020) to localize objects in images, with VL-SAM further refining the masks using SAM. Both methods identify all objects by enumerating categories during MLLM decoding. In contrast, Kang et al. (2025a) proposed a method that selects specific attention heads responsible for localization, enabling direct grounding of the object referred to by the given expression. However, this head-selection method shows poor generalization: attention heads identified on referring datasets transfer poorly and yield low accuracy on reasoning-intensive datasets.

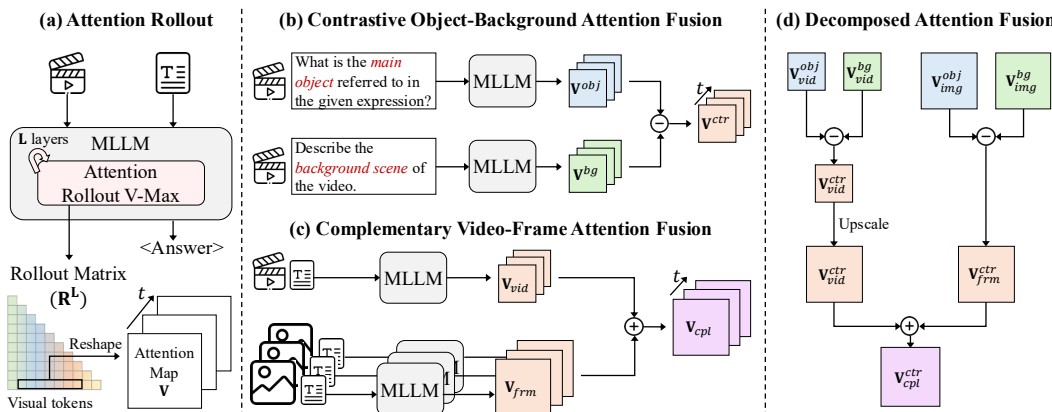

Figure 2: Overview of DecAF. **(a)** Attention rollout with our V-Max normalization produces a rollout matrix that accumulates attention across layers, from which visual-token scores for the final query token are extracted as attention maps for grounding. **(b)** Contrastive fusion suppresses attention scores on background regions. **(c)** Complementary fusion integrates video- and frame-level cues. **(d)** These fusion methods are combined into the full pipeline to refine noisy attention maps.

## 3 METHODOLOGY

### 3.1 OVERVIEW

Given a video and a text instruction referring to an object, our framework produces segmentation masks of the target object(s). The pipeline consists of two stages. First, coarse segmentation masks are obtained from attention score maps computed in an MLLM. Second, fine-grained dense segmentation masks are generated using SAM conditioned on these attention maps. In the first stage, we propose Decoupled Attention Fusion (DecAF), illustrated in Fig. 2, which integrates contrastive and complementary fusion strategies with tailored prompting methods. To obtain the attention scores, we adopt attention rollout (Abnar & Zuidema, 2020) with a new normalization technique designed for MLLMs. In the second stage, we introduce a training-free SAM2 prompting pipeline guided by attention maps (Fig. 3). Point queries are first selected by thresholding the attention maps, and SAM2 generates mask tracklets for each query. These tracklets are then evaluated with the proposed attention consistency score, which measures whether the predicted masks consistently overlap with high-attention regions across frames. The resulting scores are used to rank and tracklet candidates.

### 3.2 ATTENTION ROLLOUT WITH VISION-AWARE NORMALIZATION

We trace the influence of visual tokens on the model's output by propagating attention scores through the transformer layers of MLLMs. To better capture language-conditioned grounding, we modify the standard attention rollout (Abnar & Zuidema, 2020) with a vision-aware normalization scheme.

**Standard rollout.** Given the attention tensor $\mathbf{A}^{(l)} \in \mathbb{R}^{h \times N \times N}$ from the $l$-th transformer layer, where $h$ is the number of heads and $N$ is the total number of tokens, the head-wise averaged attention matrix is computed as Eq. 1, and the residual connection is incorporated by adding the identity matrix as Eq. 2. This reflects that a token can either propagate its own representation through the skip connection or attend to other tokens via the attention mechanism. The rollout matrix is then recursively accumulated across layers as Eq. 3, starting from the initialization $\mathbf{R}^{(1)} = \hat{\mathbf{A}}^{(1)}$, and producing $\mathbf{R}^{(L)}$, which encodes how information flows from each token to every other token throughout the network.

$$\bar{\mathbf{A}}^{(l)} = \frac{1}{h}\sum_{i=1}^{h}\mathbf{A}_i^{(l)}. \quad (1) \qquad \hat{\mathbf{A}}^{(l)} = (\bar{\mathbf{A}}^{(l)} + \mathbf{I})/2. \quad (2) \qquad \mathbf{R}^{(l)} = \hat{\mathbf{A}}^{(l)}\mathbf{R}^{(l-1)}. \quad (3)$$

**Head-wise weighted aggregation.** To reduce the effect of noisy heads, we assign a weight to each head based on the strength of its vision attention. For each layer $l$, let the original attention tensor

before aggregation be denoted as $\mathbf{A}^{(l)} \in \mathbb{R}^{h \times N \times (N_v + N_t)}$, where $N_v$ and $N_t$ indicate the number of visual and textual tokens, respectively. From $\mathbf{A}^{(l)}$, the vision block is extracted: $\mathbf{A}_v^{(l)} \in \mathbb{R}^{h \times N \times N_v}$. The maximum value over the visual token dimension is then computed as:

$$\mathbf{m}^{(l)} = \max_{j=1}^{N_v} \mathbf{A}_v^{(l)}[:, :, j], \quad \mathbf{m}^{(l)} \in \mathbb{R}^{h \times N}. \tag{4}$$

Averaging $\mathbf{m}^{(l)}$ over the token dimension finally produces the head-wise weight vector, $\mathbf{w}^{(l)} \in \mathbb{R}^h$. The weights are normalized so that $\max_h(w_h^{(l)}) = 1$, and these normalized weights are used to aggregate the heads, resulting in the final attention weights $\hat{\mathbf{A}}^{(l)} \in \mathbb{R}^{N \times (N_v + N_t)}$.

## 3.3 DECOMPOSED ATTENTION FUSION

The attention rollout mechanism quantifies token-to-token influence. To perform text-conditioned video reasoning segmentation, we cast the task as video question answering, where the goal is to identify a category of the object in a video referred to by the text instruction. We then exploit the rollout matrix values with the last token as query and visual tokens as keys, using them as attention scores that indicate how visual tokens contribute to answering the video QA, as shown in Fig. 2 (a).

However, the rollout matrix aggregates signals across all heads and layers and is too noisy to serve directly as a segmentation score map. In addition to pervasive noise, we observe strong activations in irrelevant regions, known as the *visual attention sink* phenomenon. To address this, we introduce *Decomposed Attention Fusion* (DecAF) to obtain cleaner, object-focused attention maps. As shown in Fig. 2 (d), DecAF applies contrastive fusion within each modality (video and frame) in parallel, followed by complementary fusion after upscaling the video-level attention maps to match the frame-level size. The resulting attention maps are then converted into coarse segmentation masks via thresholding. Here, we explain with shortened prompts, but the full prompts are in the Appendix.

**Contrastive Object–Background Attention Fusion.** A key challenge of using attention maps for segmentation is that irrelevant regions often receive very high scores, which cannot be suppressed by simple thresholding. Such *visual attention sinks* frequently appear regardless of the given instruction. To address this issue, we introduce *contrastive fusion*, which contrasts attention maps obtained from object-focused and background-focused prompts. Subtracting background from object attention effectively highlights the target region while suppressing spurious responses.

The specific process follows Fig. 2 (b). The object attention map is obtained by prompting the model to identify the target object category from the referring expression using an object-focused prompt template, *"What is the main object referred to in the given expression?"* The rollout attention weights from this response form the positive map. For the background attention map, we first use a background-focused prompt such as *"Describe the background scene of the video."* However, this may cause the target object to be mistakenly attended when it is not the main salient object but still appears in the background. To mitigate this, we additionally insert the identified category $o_{\text{name}}$ into the template, to explicitly exclude the target object from the background attention map. The rollout attention map from this response serves as the negative map.

Both object and background attention maps are reshaped into $(T, H_p, W_p)$, where $T$ is the number of frames and $(H_p, W_p)$ is the patch grid. Before fusion, Gaussian smoothing is applied to both maps to mitigate the sparsity of raw attention weights. The contrastive map, $\mathbf{V}^{ctr}$, is then computed by subtracting the background map from the object map, clamped to remove negative values. Finally, min–max normalization is applied to scale the values into the $[0, 1]$ range.

**Complementary Video–Frame Attention Fusion.** The softmax operation in attention enforces that all token scores sum to one. With video inputs, this constraint spreads attention across a large number of tokens, yielding maps that are relatively sparse and shaped by temporal context. In contrast, with image inputs, attention is concentrated on fewer tokens and tends to emphasize object-centric spatial details. We therefore exploit these complementary properties of video- and frame-level attention maps to achieve more robust localization.

As shown in Fig. 2 (c), we apply the identical attention rollout pipeline individually to the video and frame modalities, where each frame in the image modality is processed along the batch axis. This mixed-modality design introduces two modifications in the contrastive fusion step. (1) Since background prompting requires an object category, we select a single prediction by aggregating out-

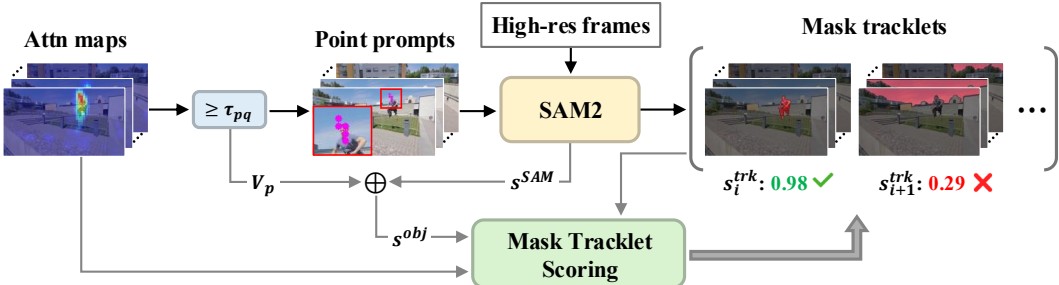

Figure 3: Overview of our SAM prompting pipeline with attention maps. (1) Point queries for SAM2 are obtained from attention maps via thresholding ($\tau_{pq}$). (2) During mask propagation, highly overlapping masks are removed. (3) Spurious mask tracklets are removed using our scoring method.

puts from both video- and frame-level inputs with object category choice prompt. (2) For min–max normalization, we normalize frame-level maps independently per frame, while video-level maps are normalized globally across all frames. Finally, the two sets of maps are fused by simple averaging, combining the global temporal context of video attention with the spatial precision of frame attention.

Our video–frame decoupled prompting enables multi-scale processing, allowing higher-resolution inputs to be used for frame attention. Recent MLLMs, such as InternVL and LLaVA-NeXT, support dynamic image resolutions via tiling, whereas video inputs remain constrained to lower resolutions (*e.g.*, 448). In contrast, QwenVL supports native resolutions for both video and image; in this case, we simply double the width and height for image inputs. To align modalities, low-res video attention maps are upsampled to match the frame-level resolution before fusion.

### 3.4 SAM2 PROMPTING WITH ATTENTION MAPS

After DecAF process, we obtain the spatio-temporal attention maps, $\mathbf{V} \in \mathbb{R}^{T_s \times H_p \times W_p}$, where $T_s$ is the number of sampled frames and $(H_p, W_p)$ is spatial resolution of visual token grid. Since this resolution is coarse, we introduce a method to prompt SAM2 using the attention maps to produce fine-grained object masks, $\hat{\mathbf{M}} \in \mathbb{R}^{T \times H \times W}$. Here, we use full frames at high-resolution, rather than the sampled frames used in the MLLM. The overall pipeline is illustrated in Fig. 3.

**Point Query Generation.** Since SAM requires spatial prompts, we generate point queries directly from attention maps to guide object mask prediction. We select visual tokens with attention scores above a threshold $\tau_{pq}$ and use their center coordinates as point queries. The set of point queries is defined as in Eq. 5, where $o_x$ and $o_y$ denote half the token width and height, respectively, ensuring that each point corresponds to the token center.

$$\mathcal{P} = \{p = (t, y + o_y, x + o_x) \mid \mathbf{V}_{t,y,x} \geq \tau_{pq}\}. \tag{5}$$

**Frame-wise Prompting and Propagation.** Starting from the first frame, the point queries are fed sequentially to SAM2 which produces frame-level masks and propagates them through subsequent frames. This process generates a video mask for each point query ($p_i$), denoted as $\mathbf{M}_i \in \mathbb{R}^{T_s \times H \times W}$, together with its confidence score, $s_i^{\text{SAM}}$, predicted by SAM2.

Naive thresholding may generate a large number of redundant masks. To reduce computation, we assign an object score $s_i^{obj} = \mathbf{V}_{p_i} + s_i^{\text{SAM}}$ for each predicted mask, where $\mathbf{V}_{p_i}$ is attention score of $p_i$. We then apply non-maximum suppression (NMS) using this object score. Two masks are considered overlapping if their IoU exceeds a threshold (*e.g.*, 0.7), and the one with the lower score is removed. If a propagated mask from previous frames highly overlaps with a new mask in the current frame, we retain only the one with the higher object score. Through this process, we obtain $K$ video mask tracklets, where $K << |\mathcal{P}|$, effectively reducing redundancy while keeping high-quality candidates.

**Mask Tracklet Scoring and Selection.** Since attention maps are at low resolution, point queries are not spatially precise and may fall on background regions. Nevertheless, SAM often produces high-confidence masks from such queries (*e.g.*, walls), leading to false positives. To suppress these

Table 1: Comparison of MLLM-based text-conditioned VOS methods that directly compute masks from attention maps (Attn Mask). All methods are training-free and grouped by MLLM.

| Method | MLLM | Ref-DAVIS | | | ReasonVOS | | | ReVOS (Overall) | | | ReVOS (Referring) | | | ReVOS (Reasoning) | | |
|---|---|---|---|---|---|---|---|---|---|---|---|---|---|---|---|---|
| | | $\mathcal{J}\&\mathcal{F}$ | $\mathcal{J}$ | $\mathcal{F}$ | $\mathcal{J}\&\mathcal{F}$ | $\mathcal{J}$ | $\mathcal{F}$ | $\mathcal{J}\&\mathcal{F}$ | $\mathcal{J}$ | $\mathcal{F}$ | $\mathcal{J}\&\mathcal{F}$ | $\mathcal{J}$ | $\mathcal{F}$ | $\mathcal{J}\&\mathcal{F}$ | $\mathcal{J}$ | $\mathcal{F}$ |
| Loc-Head [CVPR'25] | LLaVA-7B | 18.9 | 23.2 | 14.5 | 12.2 | 14.1 | 10.2 | 12.6 | 15.0 | 10.2 | 14.1 | 17.4 | 10.8 | 11.1 | 12.7 | 9.5 |
| Loc-Head [CVPR'25] | LLaVA-OV-7B | 15.9 | 14.1 | 17.7 | 12.3 | 11.2 | 13.4 | 13.1 | 12.0 | 14.3 | 14.9 | 13.8 | 16.0 | 11.4 | 10.3 | 12.5 |
| DecAF [Ours] | LLaVA-OV-7B | 21.6 | 24.0 | 19.1 | 17.2 | 19.4 | 15.0 | 15.6 | 17.6 | 13.7 | 16.9 | 19.5 | 14.4 | 14.3 | 15.6 | 12.9 |
| Loc-Head [CVPR'25] | InternVL3-8B | 19.0 | 24.1 | 14.0 | 14.1 | 15.6 | 12.6 | 14.6 | 16.8 | 12.4 | 16.5 | 19.3 | 13.7 | 12.7 | 14.2 | 11.1 |
| DecAF [Ours] | InternVL3-8B | 20.7 | 26.0 | 15.3 | 18.4 | 21.8 | 14.9 | 16.7 | 20.4 | 13.0 | 18.2 | 22.5 | 13.9 | 15.2 | 18.3 | 12.1 |
| TAM [ICCV'25] | Qwen2VL-7B | 2.5 | 1.8 | 3.3 | 2.8 | 2.8 | 2.9 | 2.8 | 2.9 | 2.6 | 2.9 | 3.1 | 2.7 | 2.7 | 2.8 | 2.6 |
| Loc-Head [CVPR'25] | Qwen2VL-7B | 18.8 | 23.8 | 13.8 | 9.0 | 11.1 | 6.8 | 13.2 | 17.5 | 8.9 | 16.5 | 22.2 | 10.7 | 10.0 | 12.9 | 7.2 |
| DecAF [Ours] | Qwen2VL-7B | 20.0 | 24.8 | 15.2 | 13.8 | 17.5 | 10.0 | 15.2 | 19.8 | 10.7 | 17.5 | 23.2 | 11.8 | 13.0 | 16.4 | 9.6 |
| TAM [ICCV'25] | Qwen2.5VL-7B | 3.5 | 2.8 | 4.3 | 3.7 | 3.4 | 3.9 | 4.0 | 4.0 | 4.0 | 4.1 | 4.1 | 4.1 | 3.8 | 3.8 | 3.8 |
| Loc-Head [CVPR'25] | Qwen2.5VL-7B | 19.1 | 24.2 | 14.0 | 10.7 | 13.1 | 8.3 | 14.1 | 18.6 | 9.6 | 16.9 | 22.7 | 11.0 | 11.4 | 14.5 | 8.3 |
| DecAF [Ours] | Qwen2.5VL-7B | 25.3 | 32.0 | 18.6 | 20.6 | 26.0 | 15.3 | 20.2 | 26.0 | 14.5 | 22.1 | 28.8 | 15.4 | 18.3 | 23.1 | 13.5 |

cases, we evaluate each mask tracklet using an attention consistency score ($s^{ac}$), which measures whether the mask consistently overlaps with high-attention regions across frames.

For each tracklet $i$, we then compute a combined tracklet score, $s_i^{trk} = \text{Avg}(\mathbf{V}_{p_i}, s_i^{\text{SAM}}, s_i^{ac})$. Tracklets with $s_i^{trk} \geq \tau_{trk}$ are retained and propagated across all video frames via SAM2 to generate the final dense segmentation masks. This procedure naturally supports both single-object and multi-object localization by treating each high-confidence query as an independent object hypothesis.

The computation of $s^{ac}$ is as follows. First, we obtain a binary mask for each frame by thresholding the attention map at its mean score $\mu_t$ (Eq. 6). Second, we assign the negative maximum attention score per frame, $\delta_t = -\max(\mathbf{V}_{t,:,:})$, to regions below $\mu_t$, (Eq. 7), penalizing low-attention areas. Finally, each mask tracklet is downampled to the attention map resolution, $\tilde{\mathbf{M}}_i \in \mathbb{R}^{T_s \times H_p \times W_p}$, and $s^{ac}$ is computed as a ratio of inner products (Eq. 8), where $\langle \cdot, \cdot \rangle$ denotes the tensor inner product.

$$\mathbf{M}_{t,y,x}^{Attn} = \begin{cases} 1, & \mathbf{V}_{t,y,x} \geq \mu_t, \\ 0, & \text{otherwise.} \end{cases} \quad (6) \quad \hat{\mathbf{V}}_{t,y,x} = \begin{cases} \mathbf{V}_{t,y,x}, & \mathbf{V}_{t,y,x} \geq \mu_t, \\ \delta_t, & \text{otherwise.} \end{cases} \quad (7) \quad s_i^{ac} = \frac{\langle \tilde{\mathbf{M}}_i, \hat{\mathbf{V}} \rangle}{\langle \mathbf{M}^{Attn}, \hat{\mathbf{V}} \rangle} \quad (8)$$

## 4 EXPERIMENTS

### 4.1 EVALUATION SETTING

**Datasets and Evaluation Metrics.** We evaluate our method on three referring VOS datasets: Ref-DAVIS (Khoreva et al., 2018), Ref-YouTube-VOS (Seo et al., 2020), and MeViS (Ding et al., 2023). In addition, we validate it on two reasoning VOS datasets: ReasonVOS (Bai et al., 2024) and ReVOS (Yan et al., 2024). Note that ReasonVOS provides only a test set and is used for zero-shot evaluation, whereas the other datasets include training data. For evaluation, we employ the standard VOS metrics: region similarity ($\mathcal{J}$), contour accuracy ($\mathcal{F}$), and their mean ($\mathcal{J}\&\mathcal{F}$).

**Implementation Details.** For mask generation directly from attention maps, we apply Otsu's adaptive thresholding method (Otsu et al., 1975). By default, attention rollout starts from the middle LLM layer (*e.g.*, 14 for 28 layers of Qwen2.5VL-7B), and SAM prompting threshold values of $\tau_{trk} = 0.8$ and $\tau_{pq} = 0.8$. We use publicly released MLLM checkpoints and the SAM2-hiera-large.

### 4.2 COMPARISON WITH EXISTING METHODS USING MLLMS

**Mask without SAM.** We evaluate segmentation masks obtained directly from MLLM attention maps using simple upscaling and thresholding, and compare them with existing methods in Tab. 1. Uniformly sampled 16 frames are used here. TAM (Li et al., 2025) performs poorly due to its strong dependence on predicted word tokens, making it unable to reliably ground expressions under our object-focused prompt. Further analysis of TAM's failure cases is provided in the Appendix. Loc-Head (Kang et al., 2025a) is also designed for text-conditioned segmentation, but operates in the image domain. Our method consistently outperforms Loc-Head across different MLLMs and datasets, with especially large margins on datasets require complex reasoning. This suggests

Table 2: Comparison of MLLM-based text-conditioned VOS methods. The upper gray rows correspond to training-based methods, while the lower colored rows correspond to training-free methods.

| Method | MLLM | Ref-DAVIS | | | ReasonVOS | | | ReVOS (Overall) | | | ReVOS (Referring) | | | ReVOS (Reasoning) | | |
|---|---|---|---|---|---|---|---|---|---|---|---|---|---|---|---|---|
| | | $\mathcal{J}\&\mathcal{F}$ | $\mathcal{J}$ | $\mathcal{F}$ | $\mathcal{J}\&\mathcal{F}$ | $\mathcal{J}$ | $\mathcal{F}$ | $\mathcal{J}\&\mathcal{F}$ | $\mathcal{J}$ | $\mathcal{F}$ | $\mathcal{J}\&\mathcal{F}$ | $\mathcal{J}$ | $\mathcal{F}$ | $\mathcal{J}\&\mathcal{F}$ | $\mathcal{J}$ | $\mathcal{F}$ |
| LISA [CVPR'24] | LLaVA-7B | 64.8 | 62.2 | 67.3 | 31.1 | 29.1 | 33.1 | 40.9 | 39.1 | 42.7 | 45.7 | 44.3 | 47.1 | 36.1 | 33.8 | 38.4 |
| VISA [ECCV'24] | ChatUniVi-7B | 69.4 | 66.3 | 72.5 | - | - | - | 46.9 | 44.9 | 49.0 | 50.9 | 49.2 | 52.6 | 43.0 | 40.6 | 45.4 |
| VideoLISA [NeurIPS'24] | LLaVA-Phi-3-V | 68.8 | 64.9 | 72.7 | 47.5 | 45.1 | 49.9 | - | - | - | - | - | - | - | - | - |
| GLUS [CVPR'25] | LLaVA-7B | - | - | - | 49.9 | 47.5 | 52.4 | 54.9 | 52.4 | 57.3 | 58.3 | 56.0 | 60.7 | 51.4 | 48.8 | 53.9 |
| VRS-HQ [CVPR'25] | ChatUniVi-7B | **76.0** | **72.6** | **79.4** | 54.9 | 52.6 | 57.3 | 59.1 | 56.6 | 61.6 | 62.1 | 59.8 | 64.5 | 56.1 | 53.5 | 58.7 |
| Veason-R1 [arxiv'25.08] | Qwen2.5VL-7B | - | - | - | **59.9** | **56.0** | **63.8** | **61.3** | **58.2** | **64.4** | **63.6** | **60.7** | **66.5** | **59.0** | **55.8** | **62.2** |
| Loc-Head [CVPR'25] | LLaVA-7B | 55.6 | 51.5 | 59.6 | 37.1 | 32.9 | 41.4 | 35.3 | 31.1 | 39.6 | 39.2 | 35.0 | 43.4 | 31.5 | 27.2 | 35.7 |
| Loc-Head [CVPR'25] | LLaVA-OV-7B | 24.2 | 21.3 | 27.1 | 32.6 | 29.7 | 35.4 | 31.7 | 28.3 | 35.1 | 32.8 | 29.2 | 36.5 | 30.6 | 27.4 | 33.7 |
| DecAF [Ours] | LLaVA-OV-7B | 59.4 | 54.8 | 64.0 | 52.8 | 49.3 | 56.3 | 40.0 | 35.8 | 44.1 | 43.4 | 39.1 | 47.6 | 36.6 | 32.6 | 40.7 |
| Loc-Head [CVPR'25] | InternVL3-8B | 66.3 | 62.4 | 70.2 | 44.3 | 41.0 | 47.5 | 43.7 | 39.9 | 47.5 | 46.7 | 42.9 | 50.6 | 40.7 | 36.9 | 44.5 |
| DecAF [Ours] | InternVL3-8B | 62.8 | 56.9 | 68.6 | 58.9 | 55.1 | 62.7 | 47.4 | 43.7 | 51.2 | 51.7 | 47.9 | 55.5 | 43.2 | 39.5 | 46.8 |
| Loc-Head [CVPR'25] | Qwen2VL-7B | 61.9 | 58.0 | 65.8 | 34.0 | 31.8 | 36.2 | 44.0 | 40.8 | 47.2 | 52.7 | 49.1 | 56.2 | 35.4 | 32.6 | 38.2 |
| DecAF [Ours] | Qwen2VL-7B | 64.1 | 59.4 | 68.9 | 52.5 | 49.0 | 56.0 | 45.3 | 41.6 | 49.0 | 52.7 | 48.9 | 56.4 | 37.9 | 34.3 | 41.5 |
| Loc-Head [CVPR'25] | Qwen2.5VL-7B | 64.6 | 60.2 | 68.9 | 41.1 | 37.9 | 44.3 | 47.0 | 43.3 | 50.7 | 53.1 | 49.3 | 56.9 | 40.8 | 37.2 | 44.4 |
| DecAF [Ours] | Qwen2.5VL-7B | **75.2** | **70.9** | **79.5** | **63.9** | **60.5** | **67.2** | **54.2** | **50.1** | **58.2** | **58.7** | **54.8** | **62.6** | **49.7** | **45.4** | **53.9** |

Table 3: Comparison on additional datasets.

| Method | MLLM | MeViS | | | Ref-YTVOS | | |
|---|---|---|---|---|---|---|---|
| | | $\mathcal{J}\&\mathcal{F}$ | $\mathcal{J}$ | $\mathcal{F}$ | $\mathcal{J}\&\mathcal{F}$ | $\mathcal{J}$ | $\mathcal{F}$ |
| VISA [ECCV'24] | Chat-UniVi-7B | 44.5 | 41.8 | 47.1 | 61.5 | 59.8 | 63.2 |
| VideoLISA [NeurIPS'24] | LLaVA-Phi-3-V | 44.4 | 41.3 | 47.6 | 63.7 | 61.7 | 65.7 |
| GLUS [CVPR'25] | LLaVA-7B | 51.3 | **48.5** | 54.2 | 67.3 | 65.5 | 69.0 |
| VRS-HQ [CVPR'25] | Chat-UniVi-7B | 50.6 | 47.6 | 53.7 | **70.4** | **68.3** | **72.5** |
| Veason-R1 [arxiv'25.08] | Qwen2.5VL-7B | **52.2** | 48.4 | **56.0** | - | - | - |
| Loc-Head [CVPR'25] | Qwen2.5VL-7B | 39.4 | 35.2 | 43.6 | 51.0 | 46.8 | 55.2 |
| DecAF [Ours] | Qwen2.5VL-7B | **48.1** | **44.0** | **52.1** | **59.9** | **56.2** | **63.5** |

that our method generalizes more effectively to reasoning-intensive scenarios, whereas localization heads rely on heuristic head selection and thus exhibit limited robustness.

Despite these relative improvements, attention maps remain very low resolution, and the resulting scores are still below those of conventional segmentation models. In particular, contour accuracy ($\mathcal{F}$) is much lower than region similarity ($\mathcal{J}$), reflecting the inability of low-resolution attention maps to capture fine-grained boundaries – opposite to the trend observed in segmentation-specialized models. These findings suggest that attention masks alone are too coarse for precise segmentation, but they provide a sufficient coarse localization signal to guide SAM prompting (Ravi et al., 2024).

**Mask with SAM.** We evaluate dense segmentation masks for all video frames using SAM2, and report the results in Tabs. 2 and 3, including both training-based and training-free methods. Loc-Head proposes its own SAM prompting method, but it is developed under a single-object assumption: the largest bounding box (bbox) is obtained using the convex hull algorithm. Also, prompting with an imprecise bbox may result in segmenting non-target objects. On Ref-DAVIS with LLaVA-7B, Loc-Head's bbox prompting achieves 30.3, whereas our prompting achieves 55.6. This large gap highlights the advantage of our prompting method; thus, we adopt for all subsequent comparisons.

In regards to training-free methods, our method outperforms Loc-Head across different MLLMs and datasets, including the additionally presented MeViS and Ref-YTVOS (Tab. 3). Although Loc-Head achieves slightly higher scores on Ref-DAVIS with InternVL3-8B, its performance drops substantially on ReasonVOS, which requires handling more complex expressions.

Compared with training-based methods, our method achieves comparable or even superior performance. On Ref-DAVIS, our method with Qwen2.5VL-7B outperforms VISA and VideoLISA by 5.8 and 6.4 $\mathcal{J}\&\mathcal{F}$, respectively. On MeViS, our method achieves 48.1, outperforming both VISA (44.5) and VideoLISA (44.4), and reaching performance close to VRS-HQ (50.6). It is worth noting that recent state-of-the-art models (GLUS, VRS-HQ, Veason-R1) leverage trained keyframe selection modules, whereas our method simply employs uniform sampling. Even with this difference, our approach surpasses all training-based methods on ReasonVOS, despite Veason-R1 additionally training the same MLLM (Qwen2.5VL) with an RL-based algorithm. This clearly manifests the effectiveness of our method.

Table 4: Ablation study of decomposed attention fusion.

(a) Object-background contrasting

| MLLM | Obj | Bg | Attn Mask | | SAM Mask | |
|---|---|---|---|---|---|---|
| | | | Ref-D | ReasV | Ref-D | ReasV |
| IVL3 | ✓ | | 12.4 | 13.2 | 50.8 | 54.7 |
| | ✓ | ✓ | **20.7** | **18.4** | **62.8** | **58.9** |
| QVL2.5 | ✓ | | 14.5 | 13.8 | 61.9 | 58.4 |
| | ✓ | ✓ | **25.3** | **20.6** | **75.2** | **63.9** |

(b) Video-frame complementing

| MLLM | Vid | Frm | Ref-D | ReasV |
|---|---|---|---|---|
| IVL3 | ✓ | | 46.0 | 50.2 |
| | | ✓ | 60.0 | 50.8 |
| | ✓ | ✓ | **62.8** | **58.9** |
| QVL2.5 | ✓ | | 65.9 | 58.6 |
| | | ✓ | 67.4 | 58.2 |
| | ✓ | ✓ | **75.2** | **63.9** |

(c) Multi-scale complementing

| MLLM | MS | Ref-D | ReasV |
|---|---|---|---|
| IVL3 | | 54.0 | 53.7 |
| | ✓ | **62.8** | **58.9** |
| QVL2.5 | | 72.4 | 60.5 |
| | ✓ | **75.2** | **63.9** |

Table 5: Ablation study of attention rollout.

(a) Rollout method

| Method | Ref-D | ReasV |
|---|---|---|
| Rollout (Abnar & Zuidema, 2020) | 68.4 | 56.8 |
| Rollout Max (Lin et al., 2024) | 72.9 | 60.9 |
| Rollout V-Max (Ours) | **75.2** | **63.9** |

(b) Starting LLM layer for rollout

| Layer index | Qwen2.5VL-7B | | InternVL3-8B | |
|---|---|---|---|---|
| | Ref-D | ReasV | Ref-D | ReasV |
| 7 (1/4) | 69.2 | 62.8 | 62.1 | 56.8 |
| 14 (2/4) | **75.2** | 63.9 | **62.8** | 58.9 |
| 21 (3/4) | 73.6 | **64.1** | 55.8 | **60.1** |

Table 6: SAM prompting threshold values.

| $\tau_{trk}$ | $\tau_{pq}$ | Ref-D | ReasV |
|---|---|---|---|
| 0.7 | 0.7 | 71.0 | 59.9 |
| 0.7 | 0.8 | 75.0 | 62.1 |
| 0.7 | 0.9 | 74.3 | 65.7 |
| 0.8 | 0.7 | 70.6 | 61.1 |
| 0.8 | 0.8 | **75.2** | 63.9 |
| 0.8 | 0.9 | 74.3 | 65.9 |
| 0.9 | 0.7 | 71.6 | 61.7 |
| 0.9 | 0.8 | 74.5 | 64.9 |
| 0.9 | 0.9 | 74.2 | **66.4** |

Table 7: Ablation study of computing attention consistency score ($s^{ac}$).

| Thresh ($\mu$) | Penalty ($\delta$) | Ref-D | ReasV |
|---|---|---|---|
| Not Use | | 60.0 | 52.9 |
| Otsu | | 68.3 | 59.6 |
| Otsu | ✓ | 67.9 | 61.1 |
| Mean | | 65.1 | 56.0 |
| Mean | ✓ | **75.2** | **63.9** |

Table 8: Evaluation on other sizes of MLLMs.

| MLLM | Size | Ref-DAVIS | | | ReasonVOS | | | ReVOS (Overall) | | | ReVOS (Referring) | | | ReVOS (Reasoning) | | |
|---|---|---|---|---|---|---|---|---|---|---|---|---|---|---|---|---|
| | | $\mathcal{J}\&\mathcal{F}$ | $\mathcal{J}$ | $\mathcal{F}$ | $\mathcal{J}\&\mathcal{F}$ | $\mathcal{J}$ | $\mathcal{F}$ | $\mathcal{J}\&\mathcal{F}$ | $\mathcal{J}$ | $\mathcal{F}$ | $\mathcal{J}\&\mathcal{F}$ | $\mathcal{J}$ | $\mathcal{F}$ | $\mathcal{J}\&\mathcal{F}$ | $\mathcal{J}$ | $\mathcal{F}$ |
| InternVL3 | 2B | 53.5 | 48.1 | 58.9 | 54.1 | 50.7 | 57.6 | 38.1 | 34.0 | 42.2 | 42.3 | 38.2 | 46.5 | 33.9 | 29.9 | 38.0 |
| | 8B | 62.8 | 56.9 | 68.6 | 58.9 | 55.1 | 62.7 | **47.4** | **43.7** | **51.2** | 51.7 | 47.9 | 55.5 | 43.2 | 39.5 | 46.8 |
| | 14B | **63.3** | **58.3** | **68.4** | **65.6** | **62.2** | **68.9** | 47.0 | 43.0 | 51.0 | 51.3 | 47.2 | 55.4 | 42.7 | 38.8 | 46.6 |
| Qwen2VL | 2B | 41.8 | 37.1 | 46.6 | 47.6 | 44.4 | 50.8 | 30.2 | 26.4 | 34.0 | 37.1 | 33.0 | 41.2 | 23.3 | 19.8 | 26.8 |
| | 7B | **64.1** | **59.4** | **68.9** | **52.5** | **49.0** | **56.0** | **45.3** | **41.6** | **49.0** | **52.7** | **48.9** | **56.4** | **37.9** | **34.3** | **41.5** |
| Qwen2.5VL | 3B | 58.1 | 53.6 | 62.6 | 52.9 | 49.6 | 56.2 | 39.7 | 35.6 | 43.8 | 46.5 | 42.1 | 50.9 | 32.9 | 29.1 | 36.7 |
| | 7B | **75.2** | **70.9** | **79.5** | **63.9** | **60.5** | **67.2** | **54.2** | **50.1** | **58.2** | **58.7** | **54.8** | **62.6** | **49.7** | **45.4** | **53.9** |

## 4.3 ABLATION STUDY

We use Qwen2.5-VL-7B (QVL2.5) and InternVL3-8B (IVL3) as models, and Ref-DAVIS (Ref-D) and ReasonVOS (ReasV) as datasets. By default, results are reported with QVL2.5 and $\mathcal{J}\&\mathcal{F}$.

**Decoupled Attention Fusion.** We evaluate the effectiveness of DecAF. First, we examine object-background contrastive fusion (Tab. 4a), which substantially improves attention mask accuracy on both referring and reasoning VOS datasets (*e.g.*, $12.4 \rightarrow 20.7$ and $14.5 \rightarrow 25.3$ on Ref-D) by suppressing the irrelevant regions. Similar improvements are also observed for SAM mask accuracy.

Second, video-frame complementary fusion (Tab. 4b) further enhances accuracy. For Qwen2.5VL, video-only and frame-only inputs yield 65.9 and 67.4 on Ref-D, respectively, whereas combining both achieves 75.2. Consistent gains are also observed on ReasV and with InternVL3.

For video-frame fusion, we adopt a multi-scale scheme that leverages higher resolution inputs at the frame level. While Qwen2.5VL supports native resolutions, InternVL and LLaVA-OV models require a fixed input size but can handle dynamic high resolution image inputs through tiling. As shown in Tab. 4c, this multi-scale fusion brings additional improvements, particularly for InternVL, whose attention map resolution is very low without tiling.

**Attention Rollout.** Tab. 5a compares our method with previous attention rollout methods. Our vision-aware head-weighted normalization further improves accuracy over the method of Lin et al. (2024). We also evaluate different LLM layers for rollout (Tab. 5b), and observe that selecting a middle layer yields the best overall performance.

**SAM Prompting.** Tab. 6 reports the results with different threshold values to filter point queries ($\tau_{pq}$) and tracklets ($\tau_{trk}$) used in the SAM prompting process. Increasing $\tau_{pq}$ helps filter out non-

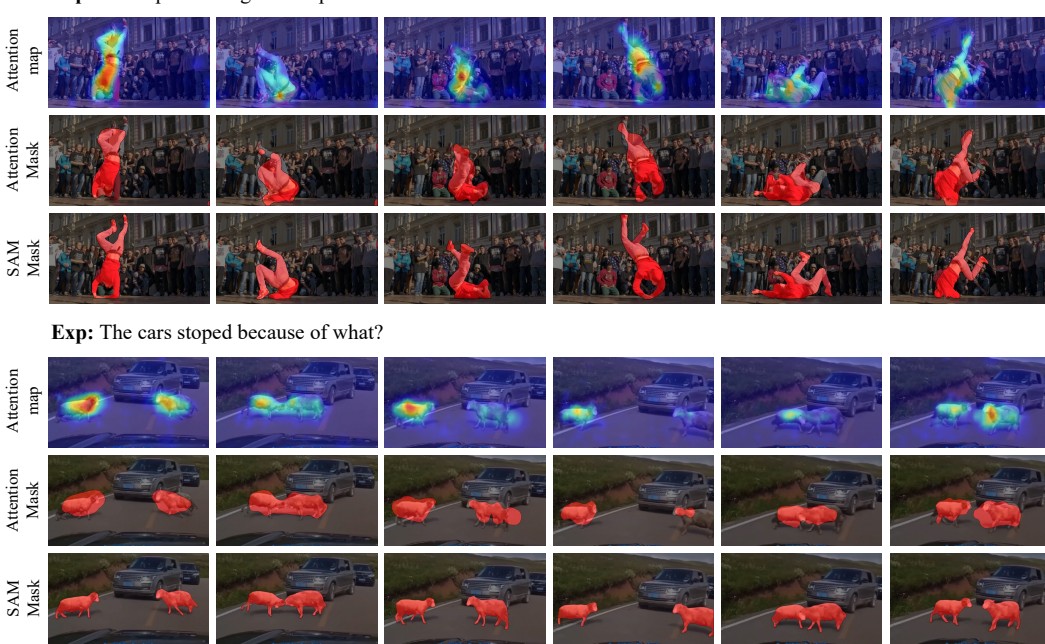

Figure 4: Qualitative results of DecAF. Visualization of refined attention maps, coarse attention masks, and dense SAM masks obtained through attention-guided prompting.

target objects or background regions, but a too high value may result in missing points. While the optimal threshold combination varies across datasets, our method remains substantially robust to threshold choices, and we use $\tau_{pq} = 0.8$ and $\tau_{trk} = 0.8$ across all datasets and models.

**Mask Tracklet Scoring.** As shown in Tab. 7, we ablate the attention consistency score ($s^{ac}$), which contributes to the mask tracklet score ($s^{trk}$). Omitting $s^{ac}$ and relying only on the object score $s^{obj}$ leads to a significant accuracy drop. For $\mathbf{M}^{Attn}$, we compare Otsu thresholding and simple averaging to obtain $\mu$, and for $\hat{\mathbf{V}}$, we evaluate both with and without the penalty term ($\delta$). Without $\delta$, Otsu thresholding yields higher accuracy than mean thresholding, as it produces tighter object masks. In contrast, with $\delta$, mean thresholding performs better, as it tends to cover the entire object region, while any included background has low attention scores and low $s^{ac}$.

**MLLM Scalability.** Tab. 8 shows that larger MLLMs generally yield better performance. InternVL3 improves from 53.5 to 63.3 on Ref-D and 54.1 → 65.6 on ReasV while Qwen2.5VL also scales effectively, with its 7B model achieving the best results across all datasets.

**Qualitative Results.** Fig. 4 illustrates the qualitative results of DecAF. Our method consistently produces refined attention maps that are precisely aligned with the referred target objects. These results demonstrate that DecAF effectively leverages the inherent localization capabilities of MLLMs without any additional training. Furthermore, our attention-guided SAM2 prompting successfully produces fine-grained dense masks. Detailed visualizations for diverse scenarios, including failure cases, are provided in the Appendix.

## 5 CONCLUSION

We explore the intrinsic localization ability of MLLMs by casting video reasoning segmentation as video QA. Based on the attention signals used for answering, we propose Decomposed Attention Fusion (DecAF), which produces robust attention maps convertible into coarse segmentation masks. To obtain fine-grained dense masks, we further introduce an attention-guided SAM2 prompting pipeline. Without training, our method surpasses training-based methods on ReasonVOS, demonstrating that MLLMs' reasoning capability directly translates into stronger localization for complex expressions. More broadly, since DecAF is agnostic to the underlying modality, it can be extended to localization tasks beyond vision – a promising direction with emerging Omni-MLLMs.

## ACKNOWLEDGMENTS

This work was partly supported by Institute of Information & communications Technology Planning & Evaluation (IITP) grant funded by the Korea government(MSIT) (No. RS-2024-00457882), the National Research Foundation of Korea (NRF) grant funded by the Korea government (MSIT) (RS-2025-00554790), and Artificial Intelligence Graduate School Program grant funded by Yonsei University (RS-2020-11201361).

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

APPENDIX

## A USE OF LARGE LANGUAGE MODELS

We utilized GPT for polishing our manuscript. Our usage is only limited to refining and grammar check of our own written draft.

## B PROMPT TEMPLATES

We describe here the prompts used to obtain object and background attention maps in Contrastive Object-Background Fusion.

**Object-focused Prompt.** The object attention map is obtained from the attention weights produced when the MLLM answers a prompt about the target object category referred to in the given expression. The prompt template is shown below:

```
{Expression}
What is the main object (or objects) referred to in the given
expression or question?
Focus on the **primary subject or agent** involved in the described
action or behavior.
Respond with a single word (e.g., 'cat', 'person', 'dog') that best
describes the target object(s).
```

**Background-focused Prompt.** In contrast, the background attention map is derived from the attention weights produced when the MLLM responds to a prompt that asks it to describe the background, excluding the target object category, in a single word or short phrase. The prompt template is shown below:

```
Describe the background scene of the video, excluding any {Object
category}.
Answer the question using a single word or phrase.
```

**Object Category Choice Prompt.** The quality of this contrastive fusion relies on the correctness of the object category. To ensure robust category selection, we first gather category predictions from both video-level and frame-level inputs and then confirm the final target category through an explicit query. The prompt template is shown below:

Here is the prompt template:

```
Given:
- Expression:  {Expression}
- Candidate object class list:  {Object category list}
Goal:  Identify the object class referred to by the expression.
Instructions:
1.  If the expression is **clear**, rely on it directly (e.g., 'a
person driving a car' → 'person').
2.  If the expression is **vague**, use the object class list to
support your decision (e.g., check frequency and plausibility).
3.  Avoid defaulting to the most frequent class unless the expression
lacks clarity.
Output the most likely referred object class - just the label.
```

The final object category is used to construct the background-focused prompts when obtaining video- and frame-level background attention maps. Importantly, the same set of prompt templates was applied across all MLLMs and datasets without any dataset-specific or model-specific modifications.

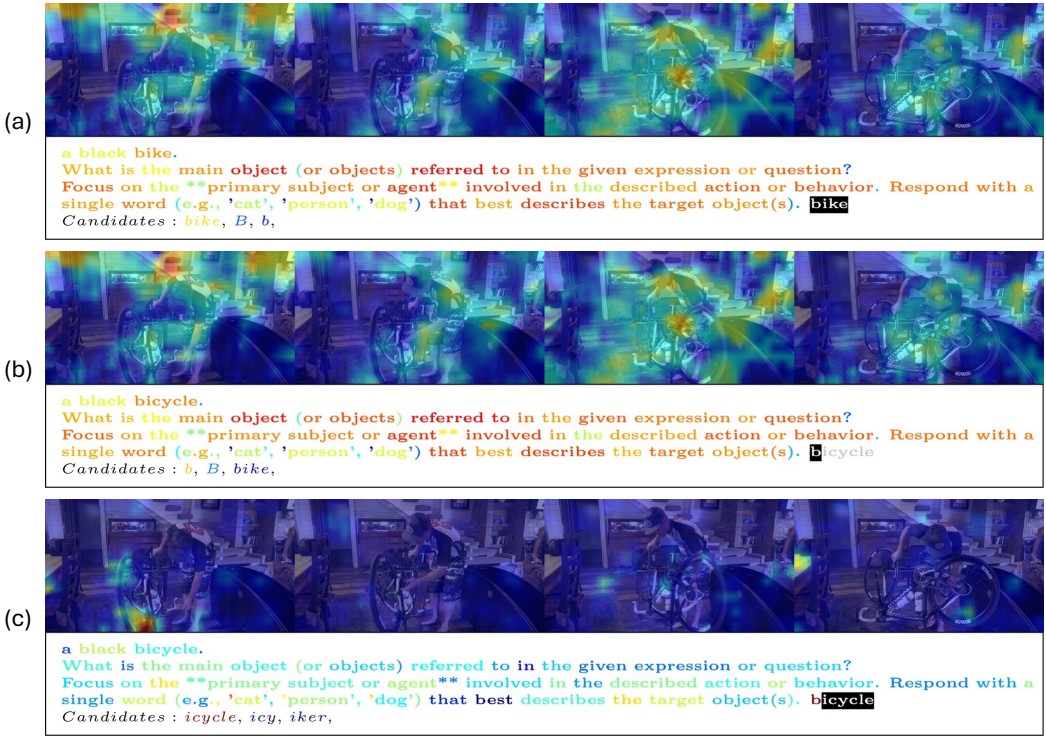

Figure 5: Analysis of TAM's failure cases

## C  MORE DETAILS ABOUT PREVIOUS METHODS

**TAM.** TAM (Li et al., 2025) exhibits strong sensitivity to the predicted word tokens. As shown in Fig. 5 (a), when the model predicts the token "bike", the resulting attention map is largely misaligned with the target object. In Fig. 5 (b), when the expression is changed to a black bicycle, the word bicycle is split into two tokens, and the first token "b" again shows severe misalignment. In contrast, Fig. 5 (c) displays the attention map for the second token "icycle", which provides a relatively better alignment with the target object. These examples demonstrate that TAM's localization is highly unstable and depends heavily on how object words are generated and tokenized.

Moreover, decoding object or background categories typically spans multiple tokens, and the original evaluation protocol reports the best-performing token (*i.e.*, the highest IoU among the predicted tokens) for each class. Such an evaluation overstates performance, underscoring TAM's lack of robustness in practical scenarios.

**Loc-Head.** Loc-Head (Kang et al., 2025a) was originally proposed in the image domain, where attention maps from MLLMs are used to segment the target object referred to by a given expression in a training-free manner. The method consists of two stages: first, identifying localization heads and then generating object masks using the attention weights from these heads.

In reproducing this method, we observed two major limitations. First, the procedure for discovering localization heads relies on sampling 1,000 image–text pairs from RefCOCO. While heads discovered from RefCOCO yielded reasonable performance when evaluated on video datasets, re-discovering heads from samples drawn directly from video datasets led to a substantial drop in performance. For example, the Attn-mask ($\mathcal{J}$) score decreased from 24.2 → 19.2 on Ref-DAVIS (Khoreva et al., 2018) and from 18.6 → 4.2 on ReVOS (Yan et al., 2024). Consequently, all experiments in our reproduction used the RefCOCO-discovered heads across datasets. Second, the head-selection process includes a heuristic that excludes heads strongly attending to the bottom row to prevent the visual attention sink phenomenon. We found that this heuristic does not generalize across all models. For example, on Qwen2VL, applying the original heuristic resulted in a score of 0.0 because the attention tended to concentrate in the right-most column rather than the bottom

row. After adapting the rule to exclude heads that strongly attend to the right-most column, the Attn-mask J improved to 23.8. Similarly, for InternVL3, enabling tiling during head discovery degraded performance, indicating further sensitivity to preprocessing choices. These results suggest that the Loc-Head procedure does not generalize reliably across either models or datasets.

A second issue arises in producing dense segmentation masks. In Loc-Head, the attention map is first binarized using the mean attention score as a threshold, after which the largest convex hull algorithm is applied to extract a bounding box. This bounding box is then used as a prompt to SAM for generating a dense mask. However, because the attention map is coarse, the resulting bounding boxes are often inaccurate, leading to large degradation in the quality of the SAM masks. When we reproduced this procedure, the performance dropped significantly compared to the paper's reported numbers; for instance, on RefCOCO validation the score decreased from $74.2 \rightarrow 34.4$. To ensure a fair comparison, we therefore applied our SAM prompting process consistently across all video datasets.

Overall, these findings highlight that the Loc-Head approach depends heavily on dataset-specific sampling, model-specific heuristics. These issues make it difficult to obtain consistent results across models and datasets. In contrast, our proposed DecAF framework works reliably across different MLLMs and datasets, providing more consistent and generalizable performance compared to the Loc-Head approach.

**Loc-Head with LLaVA-OV Details.** We attempted the following implementations for adapting Loc-Head to LLaVA-OV-7B, but Loc-Head still performs poorly with LLaVA-OV-7B, highlighting its limited robustness across models. 1. Adapting Loc-Head for tiling. LLaVA-OV-7B employs tiling to process high-resolution images. As we observed with InternVL3, extracting localization heads from tiled inputs leads to severe performance degradation. Following this observation, we disable tiling when extracting localization heads and only enable tiling during the computation of attention maps for segmentation. 2. Identifying the appropriate attention-sink region to exclude. Loc-Head removes heads with strong attention in the bottom row. Although this details is not described in the Loc-Head paper, it is implemented in the official code repository [1]. However, this heuristic does not generalize across different MLLMs. For LLaVA-OV-7B, we found that additionally excluding the left-most column is necessary for the method to produce reasonable results.

# D  QUALITATIVE RESULTS

We provide qualitative results to demonstrate the effectiveness of our proposed Decomposed Attention Fusion (DecAF) and SAM prompting. Fig. 4, 6, 7, 8 present diverse cases, including single-object, multi-object, small-object, temporal reasoning, and world knowledge scenarios. Each example shows the attention maps obtained through DecAF, the attention masks directly generated from the fused attention maps, and the dense masks obtained via SAM prompting.

Across these scenarios, DecAF consistently produces attention maps that align with instruction-referred target objects, and both the attention masks ans SAM masks accurately capture the object regions. Even in challenging settings involving multiple objects or small targets, our approach maintains robust localization and segmentation quality. Moreover, for cases requiring temporal reasoning or world knowledge, DecAF effectively leverages the capabilities of MLLMs to generate accurate masks without additional training.

We also report several failure cases (Fig. 9). As shown in Fig. 9 (a), our proposed attention consistency scoring method may underperform when the target object occupies a large area in certain frames but the attention weights cover only a small portion of that region. In such cases, the method assigns a strong penalty, leading to low scores even when the mask tracklet is correctly generated. Similarly (Fig. 9 (c)), when the target object is small and appears only briefly in the video, it occupies only a small fraction of the overall attention weights in the video, which results in low attention consistency scores and false filtering. Finally, Fig. 9 (b) shows that when the target object is extremely thin or elongated (*e.g.*, paraglider lines), the attention maps fail to capture its structure, resulting in poor masks.

---

[1] Link to the official code line for bottom-row exclusion

**Exp:** the airplane(s) flying side by side in the sky.

Figure 6: Qualitative results for the small object case.

**Exp:** Who served the ball this round?

Figure 7: Qualitative results for the temporal reasoning case.

Table 9: Comparison with naive baselines that directly use the spatial grounding of Qwen2.5-VL-7B together with SAM2. The baselines differ in how Qwen2.5-VL grounding is applied across the video: All frames performs frame-wise grounding and segmentation, First frame grounds only the first frame and propagates with SAM2, Ref & key frames uses 16 reference frames to identify the target and grounds a key frame for propagation, and 16 frames grounds uniformly sampled 16 frames. We report results on Ref-DAVIS and ReasonVOS. QVL2.5 denotes Qwen2.5-VL-7B.

| Method | Sampling | Ref-DAVIS | | | ReasonVOS | | |
|---|---|---|---|---|---|---|---|
| | | $\mathcal{J}\&\mathcal{F}$ | $\mathcal{J}$ | $\mathcal{F}$ | $\mathcal{J}\&\mathcal{F}$ | $\mathcal{J}$ | $\mathcal{F}$ |
| QVL2.5 + SAM2 | All frames | 53.0 | 50.1 | 55.9 | 44.2 | 41.9 | 46.4 |
| QVL2.5 + SAM2 | Fisrt frame | 52.8 | 50.4 | 55.2 | 38.9 | 36.7 | 41.2 |
| QVL2.5 + SAM2 | Ref & key frames | 36.5 | 33.0 | 40.0 | 23.5 | 21.8 | 25.2 |
| QVL2.5 + SAM2 | 16 frames | 64.8 | 58.0 | 71.6 | 48.0 | 41.7 | 54.3 |
| Loc-Head | 16 frames | 64.6 | 60.2 | 68.9 | 41.1 | 37.9 | 44.3 |
| DecAF [Ours] | 16 frames | **75.2** | **70.9** | **79.5** | **63.9** | **60.5** | **67.2** |

# E    NATIVE GROUNDING OF QWEN2.5VL

Tab. 9 presents the results of evaluating video segmentation using the native spatial grounding capability of Qwen2.5-VL-7B (QVL2.5) in a keypoint-prompting form together with SAM2. In all settings, QVL2.5 is directly prompted to output the target object's keypoint based on the expression, and SAM2 utilizes this keypoint for segmentation or propagation. Experiments are conducted on both Ref-DAVIS and ReasonVOS, and the details of the evaluation settings are described below.

**All frames (frame-wise grounding + per-frame SAM2).** We apply QVL2.5 independently to every frame to obtain the keypoints of target objects for that frame. Each point is then used to prompt SAM2, producing a segmentation mask for the corresponding frame without temporal propagation.

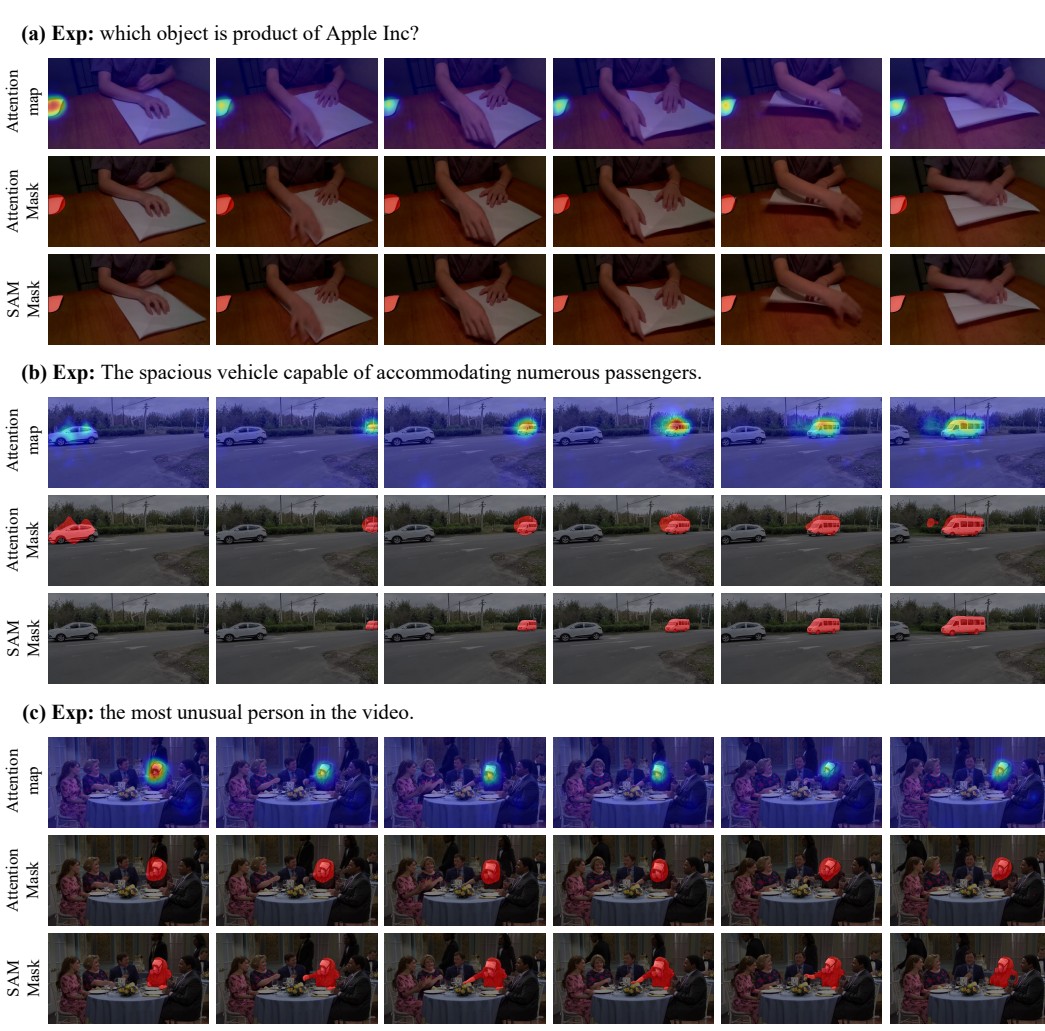

Figure 8: Qualitative results for the world knowledge cases.

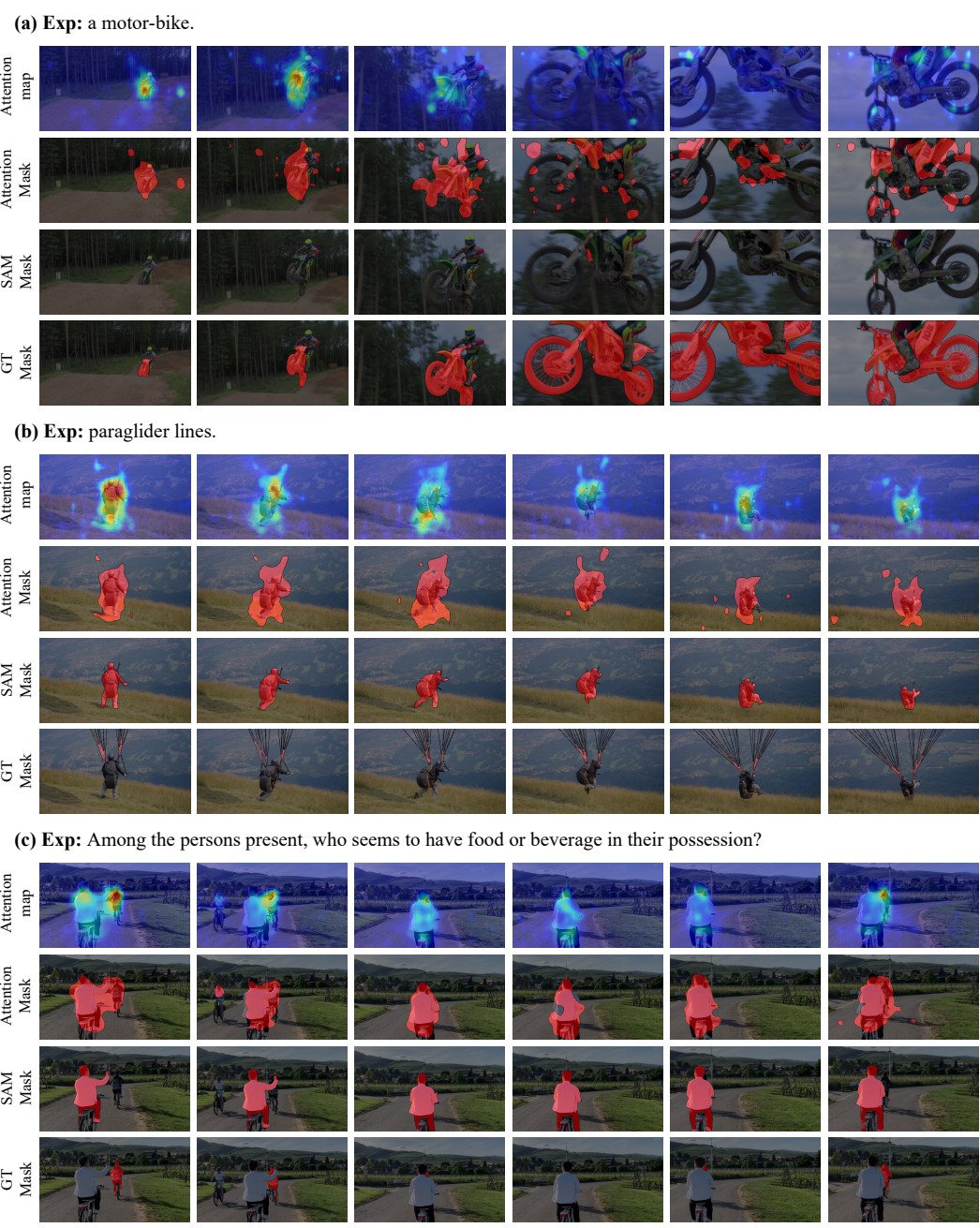

Figure 9: Qualitative examples of failure cases.

Table 10: Ablation study of the object-focused prompts.

| Prompt Design | Ref-DAVIS | | | ReasonVOS | | | ReVOS (Overall) | | | ReVOS (Referring) | | | ReVOS (Reasoning) | | |
|---|---|---|---|---|---|---|---|---|---|---|---|---|---|---|---|
| | $\mathcal{J}\&\mathcal{F}$ | $\mathcal{J}$ | $\mathcal{F}$ | $\mathcal{J}\&\mathcal{F}$ | $\mathcal{J}$ | $\mathcal{F}$ | $\mathcal{J}\&\mathcal{F}$ | $\mathcal{J}$ | $\mathcal{F}$ | $\mathcal{J}\&\mathcal{F}$ | $\mathcal{J}$ | $\mathcal{F}$ | $\mathcal{J}\&\mathcal{F}$ | $\mathcal{J}$ | $\mathcal{F}$ |
| Original | 75.2 | 70.9 | 79.5 | 63.9 | 60.5 | 67.2 | 54.2 | 50.1 | 58.2 | 58.7 | 54.8 | 62.6 | 49.7 | 45.4 | 53.9 |
| V1 (Single Sentence) | 69.8 | 65.0 | 74.7 | 63.5 | 60.3 | 66.7 | **56.3** | **52.3** | 60.2 | 60.7 | 56.8 | 64.5 | **51.9** | **47.8** | **55.9** |
| V2 (Rephrased) | **75.9** | **71.5** | **80.4** | **64.4** | **61.1** | **67.8** | 56.3 | 52.2 | **60.3** | **60.8** | **56.9** | **64.7** | 51.8 | 47.6 | 55.9 |
| V3 (Short) | 74.2 | 69.7 | 78.6 | 63.8 | 60.4 | 67.1 | 54.9 | 51.0 | 58.8 | 59.8 | 55.9 | 63.6 | 50.1 | 46.0 | 54.1 |

**First frame (single-frame grounding + SAM2 propagation).** We extract the key points only from the first frame using QVL2.5, and then prompt SAM2 to propagate the mask over the full video.

**Ref & key frames (video-conditioned grounding + SAM2 propagation).** Since QVL2.5 does not support video-level spatial grounding, we provide 16 uniformly sampled reference frames so that the model can identify the target object using the spatio-temporal context. Using this inferred target information, QVL2.5 localizes the object on the key frame–defined as the first frame among the reference frames–and extracts the corresponding keypoints. SAM2 then propagates this keypoint across the entire video to obtain the final segmentation.

**16 frames (QVL2.5 grounding + our SAM2 pipeline).** In this setting, we uniformly sample 16 frames from each video and use QVL2.5 to extract a keypoint on each sampled frame. These keypoints are then fed into our SAM2 prompting and propagation process to obtain the final video segmentation masks. For reference, in the same 16-frame setting, Loc-Head derives the keypoints from its attention maps, while DecAF uses its fused attention to obtain them.

Across all settings, native grounding with QVL2.5 shows reasonable performance but consistently remains below that of our method. The 'All frames' and 'First frame' setups rely on frame-wise grounding and therefore cannot incorporate temporal information, leading to limited accuracy. Although the 'Ref & key frames' setting attempts to provide temporal context through reference frames, QVL2.5 does not support video-level visual grounding and fails to reliably extract target keypoints under this prompting scheme.

For a fair comparison, we use the same uniform 16-frame sampling and apply the identical SAM2 prompting and propagation process as in our method. QVL2.5 grounding performs better than Loc-Head but still falls significantly short of DecAF. These results indicate that, despite not supporting video grounding natively, DecAF effectively leverages the MLLM's video understanding capability to perform robust target localization. This demonstrates that our approach, while simple, provides an effective solution for video object grounding

## F  PROMPT ROBUSTNESS ANALYSIS

We evaluate the robustness of DecAF to prompt variations by using multiple formulations of the three prompts in our pipeline: the object-focused prompt, background-focused prompt, and object category choice prompt. For each prompt type, we generate several alternative versions using Chat-GPT and evaluate how these variations affect the model's performance.

**Object-focused Prompt.** This prompt identifies the target object described in the expression. As shown in Tab. 10, we evaluate three variants: a single sentence prompt (V1), a slightly modified version of our original prompt (V2), and a prompt that is shorter than the original (V3). Overall, the three versions produce similar performance. Although V1's performance drops slightly on Ref-DAVIS, it slightly improves performance on ReVOS. Despite this small variation, the results indicate that DecAF robustly handles changes in object-focused prompt. The prompt templates are shown below:

Table 11: Ablation study of the background-focused prompts.

| Prompt Design | Ref-DAVIS | | | ReasonVOS | | |
|---|---|---|---|---|---|---|
| | $\mathcal{J}\&\mathcal{F}$ | $\mathcal{J}$ | $\mathcal{F}$ | $\mathcal{J}\&\mathcal{F}$ | $\mathcal{J}$ | $\mathcal{F}$ |
| Original | 75.2 | 70.9 | 79.5 | **63.9** | **60.5** | **67.2** |
| V1 (No Object Category) | 72.0 | 67.4 | 76.6 | 63.5 | 60.3 | 66.8 |
| V2 (Single Sentence) | **75.6** | **71.2** | **80.1** | 62.6 | 58.9 | 66.2 |
| V3 (Expression-only) | 75.5 | 71.0 | 79.9 | 63.6 | 60.1 | 67.0 |

Table 12: Ablation study of the object category choice prompts.

| Prompt Design | Ref-DAVIS | | | ReasonVOS | | |
|---|---|---|---|---|---|---|
| | $\mathcal{J}\&\mathcal{F}$ | $\mathcal{J}$ | $\mathcal{F}$ | $\mathcal{J}\&\mathcal{F}$ | $\mathcal{J}$ | $\mathcal{F}$ |
| Original | **75.2** | **70.9** | **79.5** | **63.9** | **60.5** | **67.2** |
| V1 (Single Sentence) | 73.1 | 68.5 | 77.7 | 62.4 | 59.0 | 65.8 |
| V2 (Short) | 75.1 | 70.8 | 79.4 | 63.7 | 60.4 | 67.0 |
| V3 (Expression-only) | 74.5 | 70.1 | 79.0 | 63.5 | 60.2 | 66.9 |

```
• V1 (Single Sentence)
{Expression}
Identify the primary object referred to in the expression and answer
with a single word.

• V2 (Rephrased)
{Expression}
Identify the primary object referred to in the expression.
Focus on the **primary subject or agent** involved in the described
action or behavior.  Respond with a single word (e.g., 'cat',
'person', 'dog') that best describes the target object(s).

• V3 (Short)
{Expression}
Determine the primary subject or agent mentioned in the expression
or question, and provide the object's label within a single word or
phrase.
```

**Background-focused Prompt.** This prompt identifies the background of the scene. In Tab. 11, we evaluate three variants: (V1) a prompt that queries the background without providing any object-class information, (V2) a single sentence prompt, and (V3) a prompt that excludes the object described in the expression. Since V1 does not supply the object class, the model may occasionally misinterpret the target object as part of the background, particularly when the target is not the primary object in the scene. As a result, V1 tends to perform slightly lower than the original version. Nevertheless, all three variants still show highly similar performance overall, indicating that DecAF remains robust to different formulations of the background prompt. The prompt templates are shown below:

```
• V1 (No Object Category)
Describe the background scene of the video.  Answer the question using
a single word or phrase.

• V2 (Single Sentence)
Describe the background of the video while excluding any {Object
category}, using a single word or short phrase.

• V3 (Expression-only)
Describe the background scene of the video, excluding the objects
referred to in the given expression or question {Expression}.  Answer
the question using a single word or phrase.
```

**Object Category Choice Prompt.** This prompt identifies the final object category using the object classes obtained from both the video- and frame-level predictions. In Tab. 12, we evaluate three variants: a single sentence prompt (V1), a prompt that is shorter than the original (V2), and a prompt that infers the category solely from the expression without providing the class from the object-focused prompt (V3). All variants yield similar performance, showing that DecAF robustly determines the object category across different prompt formulations. The prompt templates are shown below:

```
• V1 (Single Sentence)
Given the expression {Expression} and the candidate object classes
{Object category list}, select the single class label that best matches
the object referred to in the expression.

• V2 (Short)
Using the expression {Expression} and the candidate object classes
{Object category list}, determine which object class the expression
refers to.
If the reference is explicit, rely on the expression; if ambiguous,
use the class list as support.
Output only the most likely object class.

• V3 (Expression-only)
Given:
- Expression:  {Expression}
Goal:  Identify the object class referred to by the expression.
(e.g., 'a person driving a car' → 'person').
Output the most likely referred object class – just the label.
```

Overall, these experiments show that DecAF is robust to prompt variations. Importantly, we do not perform any prompt tuning for different MLLMs or benchmarks; all experiments in both the main paper and the appendix use the same fixed set of prompts. The consistent results across diverse prompt formulations further demonstrate that DecAF does not rely heavily on the exact choice of prompt and remains stable even when the prompts are varied.

## G  ADDITIONAL RESULTS OF FIGURE 1

Fig. 10 presents additional visualization of Fig. 1 (b), including the final fused attention maps, the attention masks obtained directly from the fused attention, the query points used for SAM2 prompting, and the resulting dense SAM masks. The fused attention maps clearly highlight strong activation in the target object. However, because the attention mask is produced via frame-wise thresholding, weak attention responses may also be converted into foreground regions. Our SAM prompting process resolves this issue by deriving query points through thresholding the fused attention map. As shown in Fig. 10, these query points emerge only within the true target region, enabling SAM2 to generate an accurate and dense segmentation mask.

## H  ANALYSIS OF SIMILAR MULTIPLE OBJECTS SCENARIO

We evaluate our method on the similar multiple objects scenario. Among the 458 samples in the ReasonVOS (Bai et al., 2024) dataset, we extract 187 samples corresponding to this challenging case. As shown in Tab. 13, the performance of the training-based method VRS-HQ (Gong et al., 2025b) decreases from 54.9 to 48.6 (-11.5%), whereas our method shows a smaller decline from 63.9 to 60.5 (-5.3%). Although performance decreases in this challenging setting, these results indicate that our approach maintains relatively stable performance when multiple similar objects are present.

We further provide qualitative comparisons with VRS-HQ in Fig. 11, 12 and 13. While VRS-HQ often produces masks on incorrect objects, our method–even with complex expressions–consistently highlights the correct target object in the attention map among multiple similar objects. This accurate localization enables our method to generate precise dense segmentation masks for the target objects.

**Exp:** Who served the ball this round?

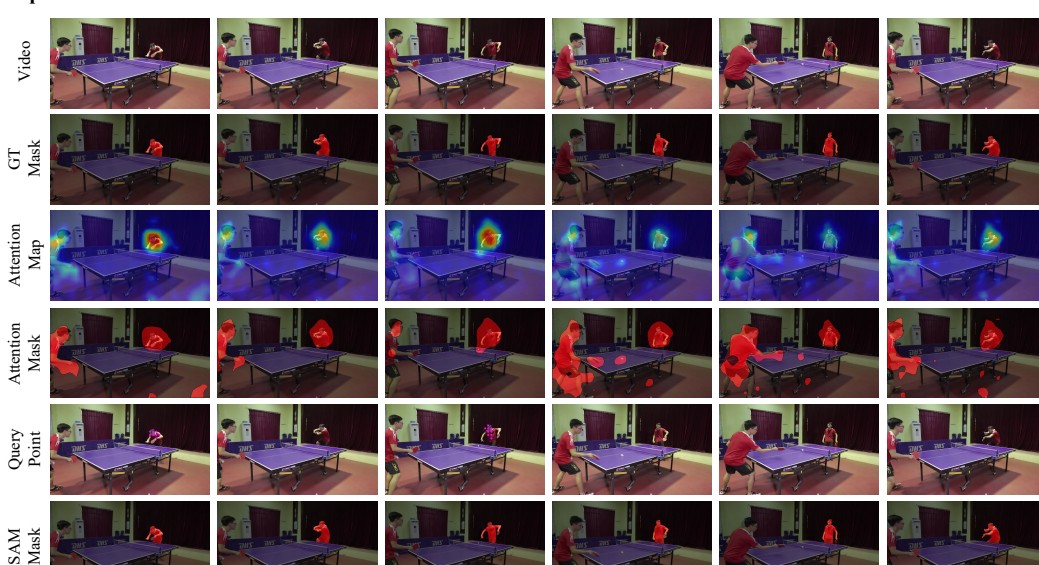

Figure 10: Additional qualitative results for Figure 1. Query points are visualized in magenta.

**Exp:** In the area around the construction site's entry point, can you identify a car positioned next to a green truck?

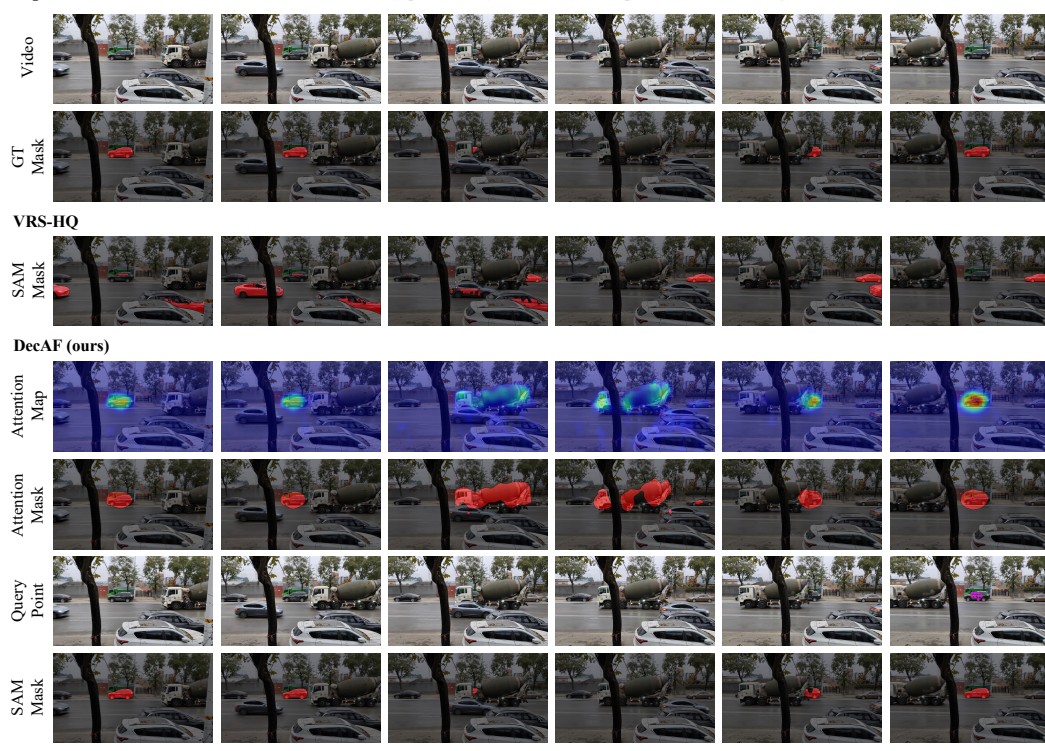

Figure 11: Qualitative results for the similar multiple objects case. Query points are visualized in magenta.

Table 13: Comparison with VRS-HQ on ReasonVOS for the similar multiple-object scenario. Fullset and Subset denote the results on all evaluation samples and the similar multiple-object samples, respectively.

| Method | Dataset Type | ReasonVOS | | |
|---|---|---|---|---|
| | | $\mathcal{J}\&\mathcal{F}$ | $\mathcal{J}$ | $\mathcal{F}$ |
| VRS-HQ [CVPR'25] | Fullset | 54.9 | 52.6 | 57.3 |
| VRS-HQ [CVPR'25] | Subset | 48.6 | 45.4 | 51.9 |
| DecAF [Ours] | Fullset | 63.9 | 60.5 | 67.2 |
| DecAF [Ours] | Subset | 60.5 | 56.4 | 64.7 |

**Exp:** Which creature seems to be keeping its distance from the rest of its kind?

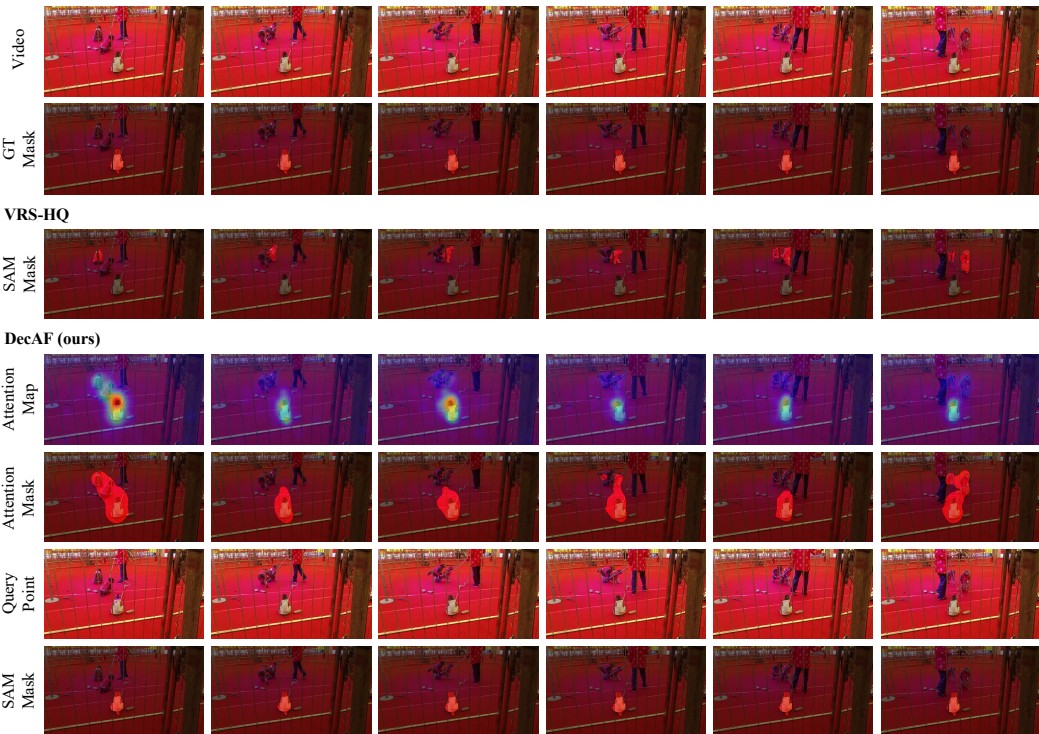

Figure 12: Qualitative results for the similar multiple objects case. Query points are visualized in magenta.

# I PART-LEVEL SEGMENTATION ANALYSIS

Our method is evaluated on object-level video segmentation since there is no video object segmentation benchmark focusing on part-level referring expressions. Nonetheless, we also examine its behavior on expressions referring to specific parts of an object, such as "the shirt of the person" or "the glasses of the person", and provide qualitative results for these part-level cases in Figs. 14, 15, 16 and 17.

Figs. 14 and 15 show that our DecAF attention maps accurately capture the regions indicated by the expression. When the expression is "the person," the attention map broadly covers the entire person, whereas for "the shirt of the person," it focuses tightly on the shirt region. As a result, both the attention mask and the corresponding query point are aligned with the shirt.

A similar trend appears in Figs. 16 and 17 for the smaller regions of "the face" and "the glasses." In both cases, the attention maps effectively highlight the intended part, and the resulting attention

**Exp:** I am driving on the road, but I have to change lanes to the right because the white vehicle in front is blocking my way. Which one is the vehicle causing me to change lanes?

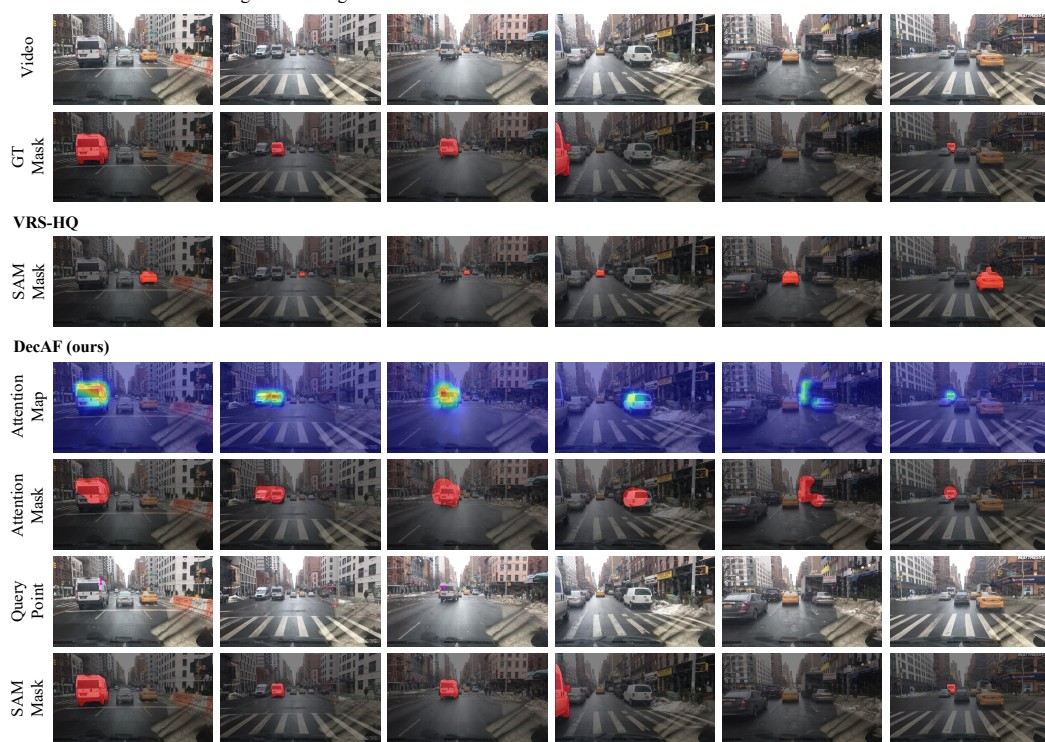

Figure 13: Qualitative results for the similar multiple objects case. Query points are visualized in magenta.

**Exp:** The person.

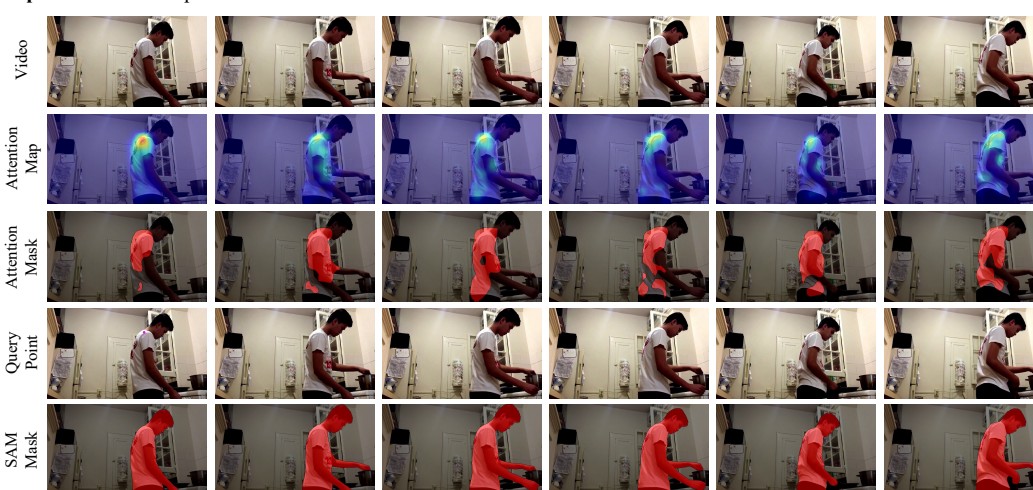

Figure 14: Qualitative results for the object-level target case (person). Query points are visualized in magenta.

**Exp:** The shirt of the person.

Figure 15: Qualitative results for the part-level target case (shirt). Query points are visualized in magenta.

masks and query points reflect this localization. Notably, the attention map for "the glasses" exhibits an even sharper focus due to the expression referring to a more specific and smaller region.

However, despite obtaining well-aligned attention maps and part-level query points, the dense masks produced by SAM2 do not consistently capture the fine-grained target regions. While the face is successfully segmented in Fig. 16, prompting SAM2 with only a single positive point can make it ambiguous whether the model should segment the object as a whole or the specific part.

Although part-level segmentation is not the primary focus of our method, these results show that DecAF is still able to perform reliable visual grounding at the part level, producing attention maps that are well aligned with the fine-grained regions referenced by the expression. Incorporating additional cues, such as negative points, could enable SAM2 to produce dense masks that precisely align with the part-level localization provided by our attention maps.

**Exp:** The face of the person.

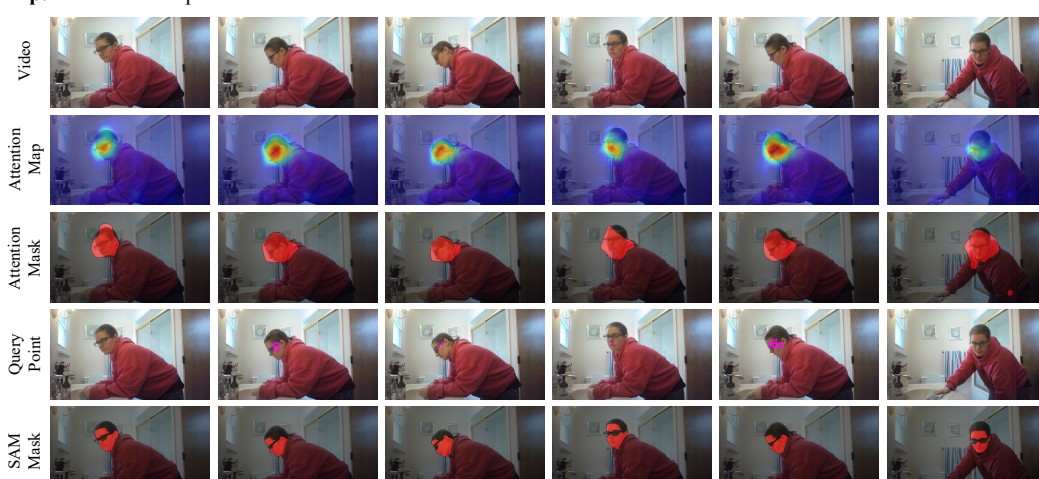

Figure 16: Qualitative results for the part-level target case (face). Query points are visualized in magenta.

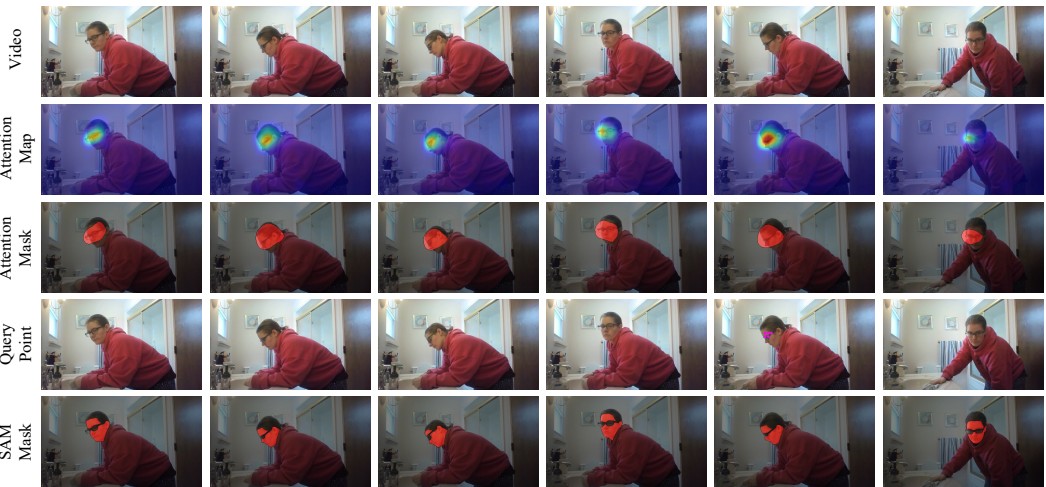

Figure 17: Qualitative results for the part-level target case (glasses). Query points are visualized in magenta.

**Exp:** Learning is an important process for self-improvement. In the scene, which object is most likely to help enhance knowledge?

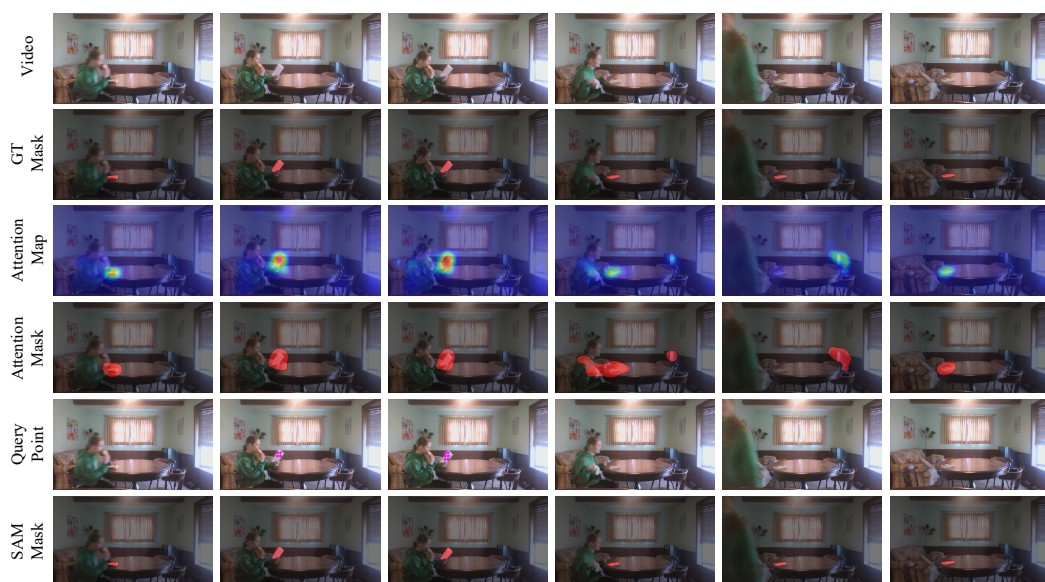

Figure 18: Qualitative results for the non-explicit object expression case. Query points are visualized in magenta.

## J    ANALYSIS OF NON-EXPLICIT OBJECT EXPRESSION CASE.

Fig. 18, 19 and 20 present qualitative results for expressions in which the target object is not explicitly described or mentioned. As shown in these examples, these expressions provide no direct visual attributes or class-level cues about the target. Instead, identifying the correct object requires reasoning over the scene context (*e.g.*, "Learning is an important process for self-improvement. In the scene, which object is most likely to help enhance knowledge?").

Our method naturally handles such challenging cases by formulating video reasoning segmentation as a Video QA task. By leveraging the MLLM's reasoning-driven attention maps–generated when answering the question formatted with the expression–this design enables the fused attention map to accurately highlight the correct target object even when the expression provides no explicit object description. Furthermore, by utilizing these well-aligned attention maps, our method can also produce accurate dense segmentation masks for the inferred target.

**Exp:** After washing hands, which item is typically used to absorb the remaining moisture?

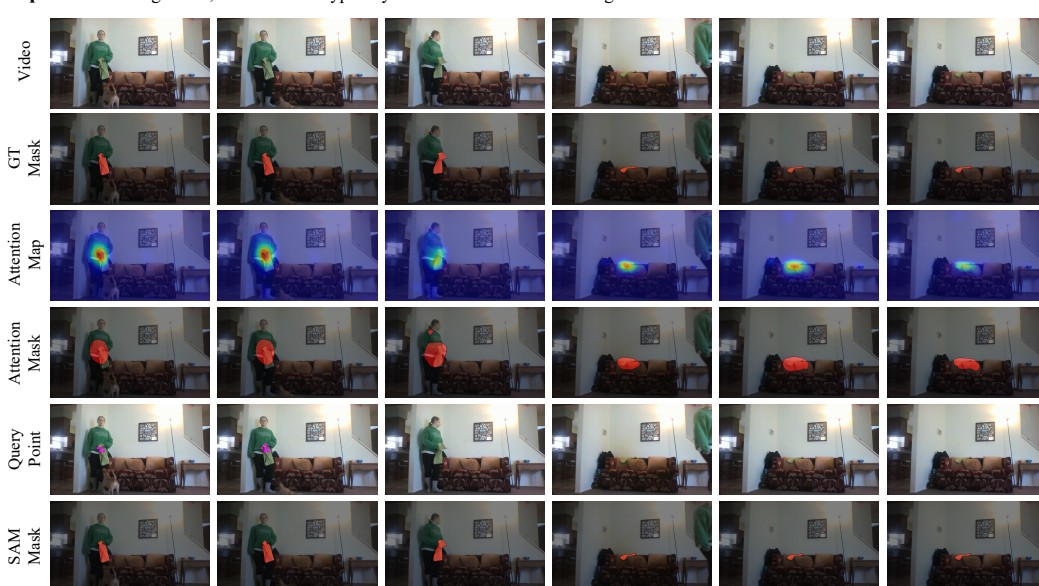

Figure 19: Qualitative results for the non-explicit object expression case. Query points are visualized in magenta.

**Exp:** The construction site has halted work due to a shortage of materials. Which object is most likely being awaited?

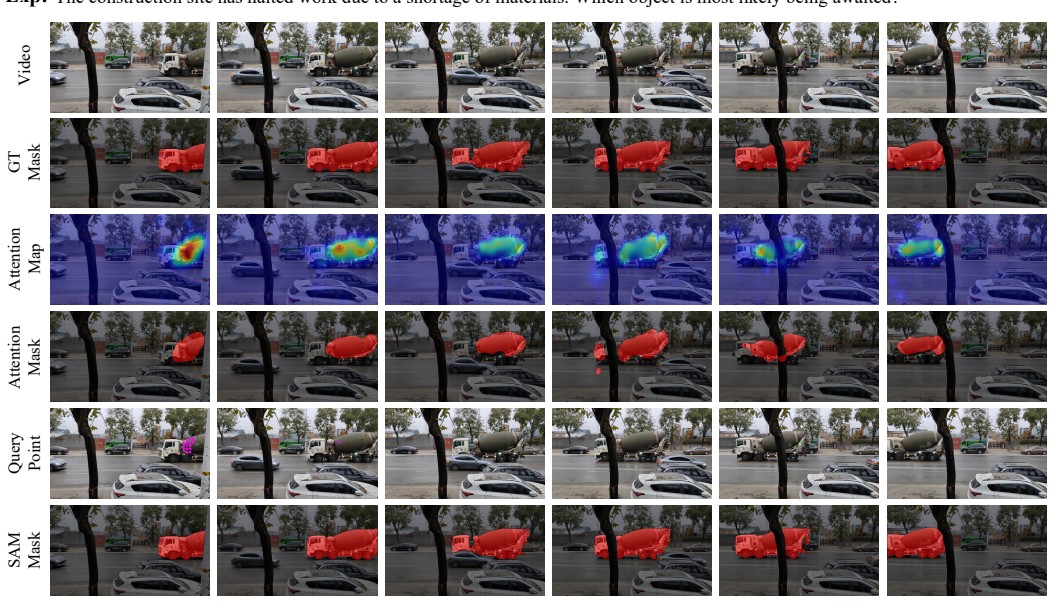

Figure 20: Qualitative results for the non-explicit object expression case. Query points are visualized in magenta.

