# OpenReview forum: "Decomposed Attention Fusion in MLLMs for Training-free Video Reasoning Segmentation"
_ICLR.cc/2026/Conference — ICLR 2026 Poster_

### Official Review · Reviewer_h74W · 2025-10-31

**Soundness:** 2
**Presentation:** 2
**Contribution:** 2
**Rating:** 4
**Confidence:** 5

**Summary:**

This paper proposes DecAF (Decomposed Attention Fusion), a training-free framework for video reasoning segmentation that leverages attention maps from Multimodal Large Language Models (MLLMs). The method introduces two key fusion strategies: (1) contrastive object-background fusion to suppress irrelevant activations, and (2) complementary video-frame fusion to combine temporal context with spatial details. The attention maps are converted to coarse masks via thresholding and refined using SAM2 with an attention consistency scoring mechanism. Experiments across five datasets demonstrate performance comparable to training-based methods.

**Strengths:**

- The decomposed attention fusion strategy is intuitive and well-designed. The contrastive object-background fusion effectively addresses the visual attention sink phenomenon, while the complementary video-frame fusion leverages the distinct strengths of different input modalities.

- The approach achieves results comparable to or better than several training-based methods (e.g., outperforming VISA and VideoLISA on Ref-DAVIS).

- Thorough Ablation Studies: The ablation studies (Tables 4-7) systematically validate each component, including fusion strategies, rollout methods, SAM prompting thresholds, and scoring mechanisms.

**Weaknesses:**

- Limited Novelty in Core Technique: While the fusion strategies are well-designed, the core building block (attention rollout) is not novel. The V-Max normalization (Eq. 4) is a relatively straightforward modification. The main contribution lies in the application and combination of existing techniques rather than fundamental algorithmic innovation.

- Computational Cost Not Discussed: The paper does not analyze computational complexity or runtime. The method requires multiple forward passes through the MLLM (for object, background, video, and frame attention maps), which could be computationally expensive. A comparison of inference time with training-based methods would strengthen the practical claims.

- Dependency on Prompt Engineering: The method heavily relies on carefully designed prompts (Appendix B). The robustness to prompt variations is not studied. Different prompt formulations might significantly affect performance, limiting generalizability.

- Missing Generalization Comparison. This is perhaps the most significant weakness of the paper. The primary advantage of training-free methods over training-based approaches is their ability to generalize to scenarios with limited or no training data. However, the paper completely fails to demonstrate this critical advantage.

**Questions:**

Please refer to the weaknesses.

---

> ### Author Response · Authors · 2025-11-25
> **Clarification of Novelty**
>
> Thank you for your valuable comments. Below please find our responses.
>
> ## W1: Limited Novelty.
>
> **(Contribution in novel formulation and complete pipeline)**
> The novelty of our work lies in the framework-level formulation that enables training-free video segmentation using existing MLLMs.
> Our key contribution is to show that video reasoning segmentation can be reformulated as a video QA task, where the MLLM's attention maps produced during question answering serve as a direct visual grounding signal.
> This use of the MLLM's internal information flow as the grounding mechanism enables training-free reasoning segmentation and, to the best of our knowledge, has not been explored in prior literature.
> Moreover, we propose a SAM2 prompting mechanism that can convert MLLM's grounding signals into dense segmentation masks to build a reliable and fully training-free video segmentation pipeline.
> We expect this complete pipeline and its component-wise analysis to serve as a strong foundation for future research on training-free adaptation of video LLMs.
>
> We would like to clarify that our main focus is not to propose a fundamentally new attention rollout algorithm.
> Instead, we introduce a straightforward yet necessary adaptation to the existing attention rollout, specifically tailored for multimodal settings to mitigate the inherent imbalance whereby visual tokens tend to receive lower attention scores.
> Despite its simplicity, this adjustment leads to non-trivial improvements (Tab. 5 (a)) and thus provides a strong baseline for future studies pursuing advanced attention rollout techniques.
> Beyond this algorithmic component, we emphasize that our complete pipeline serves as a unified grounding-evaluation framework; it allows researchers to plug in alternative approaches - not limited to attention rollout algorithms.
> More importantly, the core technical contribution of our method is decomposed attention fusion, which is an under-explored methodology; the absence of contrastive fusion leads to a significant performance drop, as shown in Tab. 4 (a).
>
> Here, we clarify the importance of each component in our method.
>
> **(Importance of V-Max normalization)**
> Vision-aware normalization, although simple, plays an essential enabling role. We observe that MLLMs exhibit an imbalance in multimodal attention distributions: visual tokens systematically receive lower attention scores than textual tokens and are suppressed when attention is recursively accumulated. Our normalization corrects this inherent issue, allowing visual attention magnitudes to remain informative during rollout. This step is necessary for any attention-based grounding method in MLLMs and is not merely an arbitrary modification.
>
> **(Importance of attention fusion approach)**
> Our attention fusion provides a generalizable solution that works consistently across different MLLMs.
> Unlike Loc-Head, which relies on model-specific heuristics that fail to generalize across MLLMs (Appendix C), our contrastive attention fusion naturally mitigates the visual attention sink phenomenon where irrelevant visual tokens receive spuriously high activation.
> Moreover, while Loc-Head is restricted to the image domain, our approach extends to the video domain without significant modification.
> Our complementary video-frame fusion effectively combines the temporal reasoning capability of video MLLMs with the fine-grained spatial cues from frame attention by leveraging different temporal and spatial resolutions.
> Our attention fusion approach demonstrates strong and stable performance across MLLMs and datasets, and we expect it to serve as a solid baseline for future research.
>
> **(Importance of SAM2 prompting method compatible with the spatio-temporal score map)**
> To complete our pipeline, we introduce an attention-guided integration with SAM2 that converts reasoning-derived grounding signals into dense spatio-temporal masks without any fine-tuning. This mechanism is fundamentally different from the typical use of SAM2, which relies on explicit visual prompts, and it provides a practical way to perform dense reasoning segmentation using activation maps produced by the MLLM's internal reasoning process, rather than being limited to the signals obtained by attention rollout.

---

> ### Author Response · Authors · 2025-11-25
> **Official Comment by Authors**
>
> ## W2: Inference Time Analysis.
>
> Please refer to our General Response, where we provide a detailed inference time analysis of our method and address this concern thoroughly.
>
> ## W3: Prompt Robustness Analysis.
>
> Our prompts were not over-tuned to any benchmark; instead, we simply used an LLM to design prompt templates that follow conventions widely adopted in previous work.
> Following the reviewer's suggestion, we conducted additional ablations using three prompt variants for each type of prompt.
> Across all cases, our method demonstrates robustness to different prompts.
>
> **Object-focused prompts.** We evaluated: (v1) a single-sentence long prompt, (v2) a rephrased prompt, and (v3) a shorter prompt.
> While v1 results in a performance drop on Ref-DAVIS, it rather improves performance on ReVOS.
> Considering that Ref-DAVIS contains only 244 video-expression pairs, whereas ReVOS 5822 pairs, the drop on Ref-DAVIS is more likely a fluctuation in the small-scale benchmark.
>
> **Table L: Ablation study of the object-focused prompts.**
> | Prompt Design | Ref-DAVIS | ReasonVOS | ReVOS |
> | ----- | ----- | ----- | ----- |
> | Original | 75.2 | 63.9 | 54.2 |
> | (v1) Single Sentence | 69.8 | 63.5 | 56.3 |
> | (v2) Rephrased | 75.9 | 64.4 | 56.3 |
> | (v3) Short | 74.2 | 63.8 | 54.9 |
>
> **Background-focused prompts.** We evaluated: (v1) a no-object category prompt, (v2) a single-sentence long prompt, and (v3) an expression-only prompt.
> Even when providing no information, neither object category nor expression, the performance drop on Ref-DAVIS is small.
> In addition, the expression-only prompt, which replaces the predicted object category with the raw expression, achieves performance similar to the original version.
> This indicates that the background-focused prompt is not sensitive to whether the object category is provided.
>
> **Table M: Ablation study of the background-focused prompts.**
> Prompt Design | Ref-DAVIS | ReasonVOS |
> | ----- | ----- | ----- |
> | Original | 75.2 | 63.9 |
> | (v1) No Object Category | 72.0 | 63.5 |
> | (v2) Sinlge Sentence | 75.6 | 62.6 |
> | (v3) Expression-only | 75.5 | 63.6 |
>
> **Object category choice prompt.** We evaluated: (v1) a single-sentence long prompt, (v2) a shorter prompt, and (v3) an expression-only prompt.
> All variants show similar performance, without a large drop.
>
> **Table N: Ablation study of the object category choice prompts.**
> Prompt Design | Ref-DAVIS | ReasonVOS |
> | ----- | ----- | ----- |
> | Original | 75.2 | 63.9 |
> | (v1) Single Sentence | 73.1 | 62.4 |
> | (v2) Short | 75.1 | 63.7 |
> | (v3) Expression-only | 74.5 | 63.5 |
>
> Full details and results are provided in Appendix F, Tabs. 10, 11, and 12.
> If additional prompt variants are of interest, we are willing to experiment with them.
>
> ## W4: Generalization with Limited or No Training Data.
>
> ReasonVOS is a zero-shot evaluation benchmark that provides no training data.
> Although the video sources overlap with existing datasets, the text expressions are newly annotated, with an emphasis on temporal reasoning.
> On this benchmark, we clearly demonstrate the generalization capability of our training-free method compared to training-based methods.
> As shown in Tab. 2, our method outperforms all training-based methods by a clear margin.
>
> VRS-HQ is trained on a large-scale multi-task data, including semantic segmentation, referring and reasoning image segmentation, image QA, referring and reasoning video segmentation, and video QA.
> Veason-R1 employs the same MLLM backbone (Qwen2.5VL-7B) as ours but further fine-tunes it using a sophisticated reinforcement learning pipeline with carefully curated chain-of-thought annotations.
> Despite these extensive training efforts, our training-free method still surpasses both methods - by **4.0 over Veason-R1 and 9.0 over VRS-HQ**.
> These results provide strong empirical evidence that our method generalizes effectively to complex reasoning-based expressions without requiring any additional training data, highlighting a key advantage of the proposed method.
>
> **Table O: Comparison on zero-shot test dataset for evaluation of generalization capability.**
> | Method | ReasonVOS |
> | ----- | ----- |
> | VRS-HQ | 54.9 |
> | Veason-R1 | 59.9 |
> | DecAF | 63.9 |

---

### Official Review · Reviewer_M2r8 · 2025-10-31

**Soundness:** 3
**Presentation:** 3
**Contribution:** 3
**Rating:** 6
**Confidence:** 4

**Summary:**

This paper proposes DecAF, a training-free framework for video reasoning segmentation. The core idea is to leverage the inherent localization abilities of Multimodal Large Language Models in the attention maps by treating the task as a video question-answering problem. To address the noise in vanilla attention maps, DecAF introduces Contrastive Object-Background Fusion and Complementary Video-Frame Fusion. The resulting fused attention map provides a strong coarse localization signal, which serves as prompt for the SAM2 model. Experiments show that DecAF significantly outperforms previous training-free methods and achieves performance comparable to SOTA training-based methods.

**Strengths:**

1. The idea of exploiting attention map of MLLM as localization signal is interesting. Although using attention map as segmentation has been widely studied in ViT models, it is interesting to see this technique to be applied in the Reasoning VOS task with MLLM as the backbone.
2. The Contrastive Object-Background Fusion and Complementary Video-Frame Fusion modules are intuitive and show compelling results.
3. The experiments and ablation studies are comprehensive.

**Weaknesses:**

1. To achieve the background contrast and video-frame fusion, it seems like the model need to run inference multiple times, which can significantly increase the cost. It is necessary to discuss this and compare with existing methods.
2. How to correctly get the background attention mask? If I understand correctly, the prompt should specify salient objects. What is the source of salient objects? Is it given by human or?
3. In reasoning segmentation, the text queries can be complex and sometimes don't explicitly mention a specific object. E.g., it can be a question that simply ask "Which object in the scene that ...?" How does this method handle such prompt?
4. The method is limited to only process object-level segmentation from the design. Is it able to segment part of object?

**Questions:**

See Weakness.

---

> ### Author Response · Authors · 2025-11-25
> **Official Comment by Authors**
>
> Thank you for your valuable comments. Below please find our responses.
>
> ## W1: Infernce Time Analysis.
>
> Please refer to our General Response, where we provide a detailed inference time analysis of our method and address this concern thoroughly.
>
> ## W2: Clarification of Background-focused Prompt.
>
> For the background-focused prompt, we use the object category predicted by the MLLM from the object-focused prompt. Please note that
> (1) no human intervention is involved at any stage, and
> (2) we are not providing external labels or salient-object information, but only object category inferred directly from the given text expression.
>
> This prompt design is motivated by the following observation. When the target object is visually similar to the background or not salient, using a generic background prompt (e.g., ``Describe the background scene of the video.'') may cause the background attention to incorrectly highlight the target object, leading to undesired suppression during contrastive attention fusion.
>
> To validate this design, we conduct an ablation on different versions of the background-focused prompt:
> (v1) no additional information, and
> (v2) using the raw referring expression instead of the predicted object category.
> As shown in the Tab. K, omitting object category information (v1) leads to a marginal drop on Ref-DAVIS, while using the expression directly (v2) maintains robust performance. Full experimental details are provided in Appendix F.
>
> **Table K: Ablation of background-focused prompt.**
> | Background Prompt Design | Ref-DAVIS | ReasonVOS |
> |--------|----------------|----------------|
> | Object Category  (Original) | 75.2 | 63.9 |
> | None (v1) | 72.0 | 63.5 |
> | Referring Expression (v2) | 75.5 | 63.6 |
>
>
> ## W3: Case of Non-Explicit Object Expression.
>
> The ReasonVOS dataset frequently includes text queries that do not explicitly mention the target object and instead require reasoning to infer the correct reference.
> For example, the expression given in Fig. 19 is "Learning is an important process for self-improvement. In the scene, which object is most likely to help enhance knowledge?".
> Even with this complex expression, the fused attention map focuses on the book being read in the scene, enabling accurate segmentation.
>
> We provide visualization examples in Figs. 19-21, where the fused attention map successfully identifies the correct object in implicit or reasoning-based queries.
> Furthermore, as shown in Tab. 2, our method achieves strong performance on ReasonVOS, outperforming training-based methods on this challenging benchmark.
> As demonstrated both quantitatively and qualitatively, our method is capable of handling implicit and complex expressions by casting video reasoning segmentation as a Video QA task and leveraging the MLLM's attention maps produced when answering the question formatted with the expression.
>
> ## W4: Part-Level Segmentation.
>
> Our method is evaluated on object-level video segmentation since there is no video object segmentation benchmark focusing on part-level referring expressions.
> However, we observe that our method can also handle part-level localization (e.g., "the face of the person").
> The fused attention naturally highlights the described subregion, enabling the extraction of part-level keypoints and masks.
>
> We provide qualitative results for these part-level cases in Figs. 15-18. As shown in the examples of "the person" (Fig. 15) vs. "the shirt of the person" (Fig. 16) and "the face" (Fig. 17) vs. "the glasses"" (Fig. 18), our attention maps accurately reflect the granularity of the referring expression. When the expression denotes a full object, the attention map broadly activates the entire region, whereas for part-level expressions, it concentrates tightly on the corresponding subregion. The resulting attention masks and query points are therefore well aligned with the intended fine-grained location.
>
> While DecAF reliably produces part-level attention maps and keypoints, generating dense part masks with SAM2 is inherently challenging when only a single positive point is provided. In cases such as "the face," SAM2 produces correct part masks, but for smaller regions (e.g., "the glasses"), the model may be uncertain whether to segment the whole object or the specific part. Although part-level segmentation is not the primary goal of our framework, these qualitative results demonstrate that DecAF can effectively localize fine-grained regions, and incorporating additional cues (e.g., negative points) may further enhance SAM2’s dense part segmentation.

---

### Official Review · Reviewer_SdCm · 2025-10-31

**Soundness:** 2
**Presentation:** 3
**Contribution:** 2
**Rating:** 2
**Confidence:** 4

**Summary:**

In this paper, the authors propose a method to use the attention map from the MLLM to extract prompts to get the masks from the SAM2 model. The authors test the performance compared to the performance by directly using the attention map (without SAM2 model). The results show that the masks generated by the SAM2 model have a good performance gain compared to the attention mask.

**Strengths:**

- The experiments demonstrate strong performance on the RefVOS benchmarks.

- The authors evaluate the method's performance across different MLLMs, showing its generalizability.

- The paper includes an ablation study on the proposed modules to validate their effectiveness.

**Weaknesses:**

- Insufficient and Missing Baselines: The paper's most significant weakness is its baseline setup. While the authors compare against training-free methods like Loc-Head, they omit the most critical and simplest baseline: using the MLLM's native grounding capabilities. Modern MLLMs, particularly Qwen2.5VL (which the authors use), are already capable of generating bounding boxes or points for a given textual expression. A much simpler and more direct training-free baseline would be to use these native grounding outputs (boxes/points) as prompts for SAM2. Given that the paper's best results come from Qwen2.5VL, this "native grounding + SAM2" baseline is essential. Without it, it is impossible to determine if the complex, multi-step DecAF process adds any real value or if it is just an overly complicated method to leverage a powerful, pre-existing capability of the MLLM.

- Computational Cost and Practicality: The "Complementary Video-Frame Fusion" module raises significant concerns about inference efficiency. As described, the method requires one MLLM pass for the entire video and then additional, separate MLLM passes for each individual frame to obtain the final result. This multi-pass design would be prohibitively slow for videos of any reasonable length, making the method impractical for real-world applications. The authors must provide a detailed analysis of the computational overhead (e.g., latency or FLOPs) compared to a single-pass approach.

- Justification of "Training-Free" Claim: The paper's "training-free" claim is questionable. In Section 3.2, the method employs "Head-wise weighted aggregation." It is unclear where these "weights" originate. If any part of the method relies on optimization or selection using a "training set," the "training-free" claim is weakened. In that case, the method should be more thoroughly compared to other training-based methods on an equal footing.

- Limited Novelty and Scope: The paper's novelty appears limited. The framework feels like a straightforward extension of image-domain attention grounding (like Loc-Head) to the video domain. The core technical contributions—the fusion modules—are not particularly novel. "Contrastive Object-Background Fusion" appears to be a simple map subtraction, and "Complementary Video-Frame Fusion" is a simple averaging of multi-scale maps. Furthermore, the main comparison method, Loc-Head, was designed for images, and the “background” contribution is not video-specific. To better demonstrate the contribution of DecAF, the authors should also evaluate its performance on standard image segmentation benchmarks (e.g., RefCOCO) to show its advantage in both domains.

- Attention Map Ambiguity: The reliance on attention maps is a potential weakness, as these maps may not always align with the intended semantic answer. This ambiguity could be particularly problematic in cases with multiple similar objects or when only part of an object is the correct answer (e.g., segmenting "the shirt of a person" from "the person"). The authors should justify how the proposed method handles such nuanced cases.

- Ablation Study Concerns: (a question, I am not very sure) A point of clarification is needed regarding the ablation study. In Table 4(a), removing the "background" module causes a performance drop far below the Loc-Head baseline. This result is counter-intuitive. Why would removing a single proposed component result in performance significantly worse than the baseline? Intuitively, a model with all proposed modules removed should perform similarly to the baseline. The authors should elaborate on this finding.


In conclusion, I am not convinced of the method's practical utility for the RefVOS task. The computational cost is unclear, and it remains uncertain whether the proposed method offers a great advantage over the much simpler, un-tested baseline of using the MLLM’s native grounding capabilities to prompt SAM2.

**Questions:**

NA

---

> ### Author Response · Authors · 2025-11-25
> **Official Comment by Authors**
>
> ## W1: Baseline with Native Grounding.
>
> We agree that evaluating the MLLM's native grounding capability is important, and we include this baseline. We thoroughly test Qwen2.5-VL's native grounding under four settings: (v1) frame-wise grounding, (v2) video-wise grounding (not supported task), (v3) first-frame grounding with SAM2 propagation,  and (v4) uniform 16-frame grounding with SAM2 propagation. Among these, the 16-frame setting (v4) performs best and is the fairest comparison, as it matches our frame sampling strategy and uses the same SAM2 prompting pipeline.
>
> As shown in Tab. G, even this strongest native grounding baseline remains far below DecAF; 64.8 vs. 75.2 for Ref-DAVIS and  48.0 vs. 63.9 for ReasonVOS. Native grounding is inherently frame-based and lacks the temporal reasoning that our fused attention extracts from the video, which explains the consistent performance gap. Full experimental details and additional comparisons are provided in Appendix E and Tab. 9.
>
> **Table G: Ablation of native grounding using Qwen2.5VL-7B with SAM2.**
> | Method | Ref-DAVIS | ReasonVOS | ReVOS
> | ------ | ----- | ----- | ----- |
> | (v1) Frame-wise Grounding | 53.0 | 44.2 | 37.5 |
> | (v2) Video-wise Grounding | 36.5 | 23.5 | 25.6 |
> | (v3) First-frame Propagation | 52.8 | 38.9 | 36.8 |
> | (v4) Uniform 16-frame Propagation | 64.8 | 48.0 | 44.9 |
> | Loc-Head | 64.6 | 41.1 | 47.0 |
> | DecAF | 75.2 | 63.9 | 54.2 |
>
> To further isolate the MLLM's grounding capability without SAM2 prompting, we additionally evaluate keypoint precision. For native grounding, we use all keypoints produced until the EOS token, while for Loc-Head and DecAF we use the top-K scoring points. We then measure whether each keypoint falls inside the ground-truth mask. As shown in Tab. H, the trend is consistent with the SAM2 results: DecAF substantially outperforms both native grounding and Loc-Head. In contrast to native grounding, DecAF also outputs a confidence score for each keypoint. This additional output enables our filtering mechanism within the SAM2 prompting pipeline, which further improves segmentation accuracy, as shown in the above Tab. ReasonVOS results (63.9 vs. 48.0).
>
> **Table H:  Evaluation of direct visual grounding performance from Qwen2.5VL-7B. Top-scoring points are extracted from the attention maps while all points generated until EOS token are used for native grounding of Qwen2.5VL. We report point-precision of top-K scoring points.**
> | Method| Ref-DAVIS | | | ReasonVOS | | |
> | ------ | ----- | ----- | ------ | ----- | ----- | ----- |
> | | PP1 | PP5 | PP10 | PP1 | PP5 | PP10 |
> | Native Grounding (v4) | | 82.2  | | | 61.8 | |
> | Loc-Head | 65.2 | 60.4 | 61.9 | 23.9 | 26.9 | 27.9 |
> | DecAF | 88.9 | 88.1 | 87.6 | 73.4 | 74.9 | 73.5 |
>
>
> ## W2: Inference Time Analysis.
>
> Please refer to our General Response, where we provide a detailed inference time analysis of our method and address this concern thoroughly.
>
> ## W3: Clarification that Our Method is Training-free.
>
> Our method is strictly training-free, and the "head-wise weighted aggregation" step does not involve any learned weights or optimization.
> Although Section 3.2 and Eq. (4) describe this process, we would like to clarify the detailed mechanism here. During inference, we first compute the attention matrices for each attention head as part of the rollout procedure. From each head's attention matrix, we extract the maximum attention value over the visual token dimension, and then average these values to obtain a per-head weight. These weights are subsequently normalized and used for aggregation. Importantly, all computations are performed on-the-fly during inference; no training data, gradient updates, or learned parameters are used at any point. Therefore, our approach remains fully and strictly training-free.

---

> ### Author Response · Authors · 2025-11-25
> **Official Comment by Authors**
>
> ## W4-1: Limited Novelty.
>
> While both Loc-Head and our method aim to perform training-free visual grounding with MLLMs, the underlying algorithms are fundamentally different. Loc-Head relies on selecting a small set of localization heads using heuristics. As addressed in Appendix C, these heuristics fail to generalize: performance drops sharply when using different datasets for head selection, and the bottom-row rule does not transfer to other MLLMs. In contrast, DecAF requires no head selection or model-specific tuning. Our framework introduces a new formulation that casts video object reasoning as a Video QA task, enabling the MLLM’s reasoning attention to serve directly as a grounding signal. This formulation is novel and allows DecAF to operate robustly across different MLLMs and datasets.
>
> Regarding the reviewer’s comment on simple subtraction: although contrastive object–background fusion is implemented via subtraction, the insight and its effect are non-trivial. MLLM attention maps contain substantial noise including visual attention sinks; fusing object- and background-focused attention removes these artifacts and produces a clean, stable grounding map. Unlike Loc-Head, our approach does not depend on which head carries spatial information and works consistently across MLLMs. We also evaluate our contrastive fusion on image segmentation benchmarks, which further confirms that the idea is effective beyond videos.
>
> Finally, Loc-Head cannot be extended to video reasoning trivially since its design is tied to single-image assumptions. DecAF introduces complementary video–frame fusion, which combines temporal understanding from video attention with fine-grained spatial cues from frame attention. This mechanism is not a trivial multi-scale averaging: it leverages the complementary strengths of video MLLMs (temporal reasoning) and frame processing (high spatial detail) - a capability absent in all prior training-free grounding methods.
>
> ## W4-2: Results of Standard Image Segmentation Benchmark.
>
> Although DecAF is designed for video reasoning segmentation, we also evaluated it on RefCOCO, RefCOCO+, RefCOCOg, and ReasonSeg.
> To ensure a fair comparison, both methods were evaluated using the same Qwen2.5-VL-7B backbone.
> Since Loc-Head's SAM prompting is not provided in the official implementation (Appendix C), we conducted evaluation using attention masks obtained by applying Otsu thresholding to each method’s attention map.
>
> Loc-Head was evaluated using localization heads discovered from RefCOCO samples as defined in the original method. DecAF, in contrast, uses no dataset-specific samples and model-specific heuristics. In the image domain, it operates only with the contrastive fusion component because complementary video-frame fusion cannot be applied.
>
> Under this setup, Loc-Head performs slightly better on RefCOCO, which is expected because it directly discovers its localization heads using samples from that dataset. However, on the other benchmarks, our method consistently achieves higher performance despite operating under more constrained conditions. Although DecAF is primarily designed for the video domain, these results indicate that it also generalizes well across all image benchmarks, even when only a subset of our framework is applied.
>
> **Table I: Image referring segmentation benchmark evaluation results.**
> | Method | RefCOCO | | | RefCOCO+ | | | RefCOCOg | | ReasonSeg | |
> |--------|---------|-|-|----------|-|-|----------|-|-----------|-|
> | | val | testA | testB | val | tsetA | tsetB | val | test | val | test |
> | Loc-Head | 40.9 | 41.0 | 39.8 | 33.1 | 35.1 | 29.4 | 33.1 | 33.4 | 19.2 | 15.3 |
> | DecAF | 37.3 | 38.6 | 35.3 | 34.4 | 36.1 | 31.5 | 35.9 | 36.0 | 21.3 | 26.3 |

---

> ### Author Response · Authors · 2025-11-25
> **Official Comment by Authors**
>
> ## W5-1: Case with Multiple Similar Objects.
>
> We specifically evaluate our method on scenarios involving multiple similar objects. Among the 458 ReasonVOS samples, we identify 187 such cases. To construct this subset, we apply the following criteria: (1) the presence of multiple objects belonging to the same category, and (2) the target object cannot be trivially distinguished from the others. As shown in Tab. 13, VRS-HQ drops from 54.9 to 48.6 (-11.5\%), while our method drops from 63.9 to 60.5 (-5.3\%), indicating substantially greater robustness in this challenging setting.
>
> Qualitative results in Figs. 12-14 further show that VRS-HQ often segments an incorrect object, whereas our fused attention consistently localizes the correct target even under complex expressions. These results demonstrate that our method remains stable and discriminative when multiple similar objects are present.
>
> **Table J: Ablation of multiple similar objects on ReasonVOS benchmark.**
> | Method | Dataset Type | ReasonVOS |
> |--------|----------------|----------------|
> | VRS-HQ | Fullset | 54.9 |
> | VRS-HQ | Subset | 48.6 |
> | DecAF | Fullset | 63.9 |
> | DecAF | Subset | 60.5 |
>
>
> ## W5-2: Part-Level Segmentation.
>
> Our method is evaluated on object-level video segmentation since there is no video object segmentation benchmark focusing on part-level referring expressions.
> However, we observe that our method can also handle part-level localization (e.g., "the face of the person").
> The fused attention naturally highlights the described subregion, enabling the extraction of part-level keypoints and masks.
>
> We provide qualitative results for these part-level cases in Figs. 15-18. As shown in the examples of "the person" (Fig. 15) vs. "the shirt of the person" (Fig. 16) and "the face" (Fig. 17) vs. "the glasses" (Fig. 18), our attention maps accurately reflect the granularity of the referring expression. When the expression denotes a full object, the attention map broadly activates the entire region, whereas for part-level expressions, it concentrates tightly on the corresponding subregion. The resulting attention masks and query points are therefore well aligned with the intended fine-grained location.
>
> While DecAF reliably produces part-level attention maps and keypoints, generating dense part masks with SAM2 is inherently challenging when only a single positive point is provided. In cases such as "the face," SAM2 produces correct part masks, but for smaller regions (e.g., "the glasses"), the model may be uncertain whether to segment the whole object or the specific part. Although part-level segmentation is not the primary goal of our framework, these qualitative results demonstrate that DecAF can effectively localize fine-grained regions, and incorporating additional cues (e.g., negative points) that may further enhance SAM2’s dense part segmentation.
>
> ## W6: Clarification of Tab. 4 (a).
>
> Loc-Head is not a baseline for our method, as its core mechanism differs fundamentally from ours. Loc-Head identifies a small set of localization heads by applying model-specific heuristics (Appendix C), which requires manually inspecting and tuning attention patterns for each MLLM. Since identifying these heuristics must be done separately for every model and relies on information unavailable in a strictly training-free setting, this process effectively overfits to the benchmark. In contrast, our method applies attention rollout over all heads to ensure generalizability across MLLMs, which inevitably introduces substantially more noisy attention maps. The background-focused module is therefore essential in our pipeline, as it suppresses this noise through contrastive attention fusion. While our rollout-only baseline using only object-focused attention is indeed noisy and underperforms relative to Loc-Head, our object-background contrastive fusion effectively mitigates this issue and ultimately outperforms Loc-Head.

---

> > ### Comment · Reviewer_SdCm · 2025-11-27
> >
> > I thank the authors for their detailed response and the additional experiments. I plan to raise my score to 6 after the discussion.
> >
> > I appreciate the comprehensive comparison with the Qwen2.5-VL native grounding baseline (W1) and the clear explanation regarding the strictly training-free nature of the proposed method (W3). The clarification in W6 regarding the comparison setup is also helpful. The additional results on standard image segmentation benchmarks (W4-2) are encouraging; they demonstrate that the proposed method achieves competitive performance compared to LocHead on image tasks, while showing greater potential and robustness in the video domain due to the proposed fusion mechanisms.
> >
> > I have one remaining question regarding the baseline definition. Based on your description in W4-1 and W6, you mention that LocHead relies on "selecting a small set of localization heads using heuristics" derived from dataset samples (e.g., RefCOCO). Does this imply that LocHead is not "training-free" (or "data-free") in the strictest sense, as it appears to require a data-driven search or calibration phase to identify these specific heads?

---

> > > ### Author Response · Authors · 2025-11-30
> > > **Clarification about Loc-Head training-free definition**
> > >
> > > Yes, your understanding is correct. While Loc-Head is an important early exploration of training-free visual grounding for MLLMs, it is not strictly data-free. As addressed in W4-1, W6, and Appendix C, Loc-Head identifies localization heads through a heuristic selection process that requires careful calibration with respect to both the model (e.g., head exclusion rules) and the data (e.g., samples used for finding heads).
> > >
> > > In contrast, our method does not rely on any such heuristics; instead, it uses our proposed decomposed attention fusion method combined with the attention rollout mechanism. This design enables strong generalization across different MLLMs and datasets. Additionally, during the discussion period, following the reviewer's request, we also showed that a partial component of our method, contrastive object-background attention fusion, is effective even in the image domain, outperforming Loc-Head on RefCOCO+, RefCOCOg, and ReasonSeg datasets.

---

### Official Review · Reviewer_eWqc · 2025-11-01

**Soundness:** 4
**Presentation:** 4
**Contribution:** 3
**Rating:** 8
**Confidence:** 3

**Summary:**

This paper investigate the training-free video referring segmentation task, which is relatively underexplored previously.  The main idea of this paper is to cast referring video segementation task as video QA task and extract attention score in video LLM to perform segmentation. This paper presents a systematic pipeline including Decomposed Attention Fusion, DeCAF, and SAM2 refinement. DeCAF mainly includes two stages: 1) object-background attention rollout subtraction, and 2) sparse video and dense frame fusion. The experimental results of this proposed pipeline are on par with training-based methods on several VOS benchmarks.

**Strengths:**

1) This paper is one of the first cohorts to investigate the scope of the training-free video referring segmentation with LLM and therefore delivers substantial novelty and insights. The paper made several efforts to work this out and the results are on par with the training-based methods, which should be appreciated.

2) This paper proposed several technical methods specifically tailored for video tasks, including video-frame attention fusion, frame-wise propagation, tracklet scoring and selection, showing that the pipeline is not a simply adopted method from image domain to video domain.

3) This paper performed several ablation studies (in Table 5~8) to validate necessities of proposed components in the pipeline, which is of great value.

**Weaknesses:**

1) In Table 1 and 2, Why DecAF is with LLaVA-OV-7B while Loc-Head is with LLaVA-7B? Could you please show the results of Loc-Head woth LLaVA-OV-7B?

2) Could the authors show the inference time of the whole pipeline of video segmentation, especially compared to the training end-to-end methods? Because it seems that the training free method is bound to include heavy pretrained methods to refine coarse attention scores. Adding the inference time analysis will increase the technical soundness of this paper.

3) In the demo in Figure 1 (b), the author claims the segementation pipeline can resolve conflicts e.g., between the server and hitting player. Could authors elaborate this mechainism more in details? Is it mainly due to frame-wise prompting & progation or mask tracjlet scoring & selection?

**Questions:**

See weaknesses.

---

> ### Author Response · Authors · 2025-11-25
> **Official Comment by Authors**
>
> Thank you for your valuable comments. Below please find our responses.
>
> ## W1: Loc-Head Results with LLaVA-OV-7B.
>
> We provide the Loc-Head results with LLaVA-OV-7B in Tabs. E and F, and we also update Tabs. 1 and 2 in the main paper, accordingly.
> Overall, Loc-Head does not generalize well with LLaVA-OV-7B; its J&F score on Ref-DAVIS is very low, and its performance across datasets is consistently worse than with LLaVA-7B.
> In contrast, our method remains strong across different MLLMs and datasets.
>
> We made our best effort to adapt Loc-Head to LLaVA-OV-7B based on the official codebase, but the performance remained poor even after applying two necessary adaptations: (1) handling compatibility with the tiling technique and (2) identifying the appropriate attention-sink region. We provide the full details in Appendix C.
>
> **Table E: LLaVA-OV-7B results of the segmentation masks computed directly from attention maps.**
> | Method | MLLM | Ref-DAVIS | ReasonVOS | ReVOS (Overall) | ReVOS (Referring) | ReVOS (Reasoning) |
> |--------|------|----------------|----------------|--------------------|----------------------|-----------------------|
> | Loc-Head | LLaVA-7B | 18.9 | 12.2 | 12.6 | 14.1 | 11.1 |
> | Loc-Head | LLaVA-OV-7B | 15.9 | 12.3 | 13.1 | 14.9 | 11.4 |
> | DecAF | LLaVA-OV-7B | **21.6** | **17.2** | **15.6** | **16.9** | **14.3** |
>
> **Table F: LLaVA-OV-7B results of the segmentation masks obtained with SAM.**
> | Method | MLLM | Ref-DAVIS | ReasonVOS | ReVOS (Overall) | ReVOS (Referring) | ReVOS (Reasoning) |
> |--------|------|----------------|----------------|--------------------|----------------------|-----------------------|
> | Loc-Head | LLaVA-7B | 55.6 | 37.1 | 35.3 | 39.2 | 31.5 |
> | Loc-Head | LLaVA-OV-7B | 24.2 | 32.6 | 31.7 | 32.8 | 30.6 |
> | DecAF | LLaVA-OV-7B | **59.4** | **52.8** | **40.0** | **43.4** | **36.6** |
>
>
> ## W2: Inference Time Analysis.
>
> Please refer to our General Response, where we provide a detailed inference time analysis of our method and address this concern thoroughly.
>
> ## W3: Clarification of Figure 1 (b).
>
> The ambiguity between the server and the hitting player is mainly resolved by our **complementary video-frame attention fusion mechanism**.
>
> As shown in Fig. 1 (b), attention maps generated from frame-only input lack temporal understanding and therefore tend to highlight any player currently hitting the ball. In contrast, attention maps generated from video input **effectively leverage temporal context and correctly focus on the player** who served the ball earlier in the round. By fusing these two signals, the fused attention map strongly activates the true server while suppressing activations on the hitting player. Consequently, when point queries are computed by thresholding with $\tau_{pq}$, the selected points correspond only to the server. We provide additional visualizations in Fig. 11 to further illustrate this behavior.
>
> Although this particular case is handled by our attention fusion mechanism, we would also like to clarify how other components contribute in other scenarios.
> During **frame-wise prompting and propagation**, when multiple SAM masks overlap, we retain the highest quality mask based on the object score ($s^{obj}$), ensuring stable per-frame mask proposals.
> Beyond this, our **mask tracklet scoring and selection** module plays a key role in filtering out false positive objects across the video.
> Using the attention consistency score ($s^{ac}$), defined in Eqs. 6–8, we select tracklets that consistently align with high-attention regions over time and suppress non-target tracklets.

---

### Author Response · Authors · 2025-11-25
**General Response - Summary of Updates**

We appreciate all reviewers for their constructive feedback, which has been invaluable in improving the clarity and completeness of our work.

## Manuscript Updates.

We updated the experimental results and manuscript based on the following corrections and additions:
1. SAM2 post-processing enabled.
The hole-filling post-processing step in SAM2 was previously disabled, even though it is part of the official setup. We have now enabled this step, which leads to consistent improvements in segmentation accuracy. Note that this is not an additional engineering trick to boost performance; rather, it is necessary to ensure a fair comparison with other methods that already employ this step.
2. Corrected ReasonVOS evaluation.
A small number of ReasonVOS samples were omitted in the earlier evaluation. After including all samples, the accuracy metrics show minor adjustments.
3. Expanded appendices.

We added detailed explanations and supporting material corresponding to the rebuttal responses in Appendix E-J, including Figs. 11-21 and Tabs. 9-13.

We are willing to update the contents addressed during the discussion in the main paper as well.

---

> ### Author Response · Authors · 2025-11-25
> **General Response - Inference Time Analysis**
>
> ## Inference Time Analysis.
>
> As noted by the reviewers, our method has relatively long latency, and we provide a detailed analysis on latency below.
> However, this work is the first to tackle the general problem of transferring an MLLM's complex reasoning process into spatial localization in a training-free manner.
> Our focus lies on achieving stable and direct visual grounding from any MLLM, independent of its native grounding support; thus, inference speed is not the primary objective.
> Importantly, this training-free grounding is not available in VRS-HQ or other training-based methods, as they rely on a finetuned SAM decoder and heavily compressed visual tokens.
> We demonstrate the effectiveness of our formulation through both keypoint precision (Tab. A) and mask quality (Tab. 1).
> Moreover, since our method does not post-train an MLLM, it preserves the MLLM's inherent general reasoning ability, leading to superior performance on the ReasonVOS dataset which is a zero-shot evaluation benchmark.
>
> **Table A: Evaluation of direct visual grounding performance from Qwen2.5VL-7B. Top-scoring points are extracted from the attention maps while all points generated until EOS token are used for native grounding of Qwen2.5VL. We report point-precision of top-K scoring points.**
> | Method| Ref-DAVIS | | | ReasonVOS | | |
> | ------ | ----- | ----- | ------ | ----- | ----- | ----- |
> | | PP1 | PP5 | PP10 | PP1 | PP5 | PP10 |
> | Native Grounding | | 82.2  | | | 61.8 | |
> | Loc-Head | 65.2 | 60.4 | 61.9 | 23.9 | 26.9 | 27.9 |
> | DecAF | 88.9 | 88.1 | 87.6 | 73.4 | 74.9 | 73.5 |
>
> Notably, DecAF is the only training-free method capable of handling video-level temporal reasoning (Fig. 11), and it consistently outperforms both Loc-Head and native Qwen2.5VL grounding across all metrics.
> Achieving this capability requires an additional MLLM pass on the video input, which complements the high-resolution frame-level passes used for fine-grained spatial cues.
> In contrast to Loc-Head, our method does not rely on model-specific heuristics (Appendix C).
> Unlike native grounding from Qwen2.5VL, our method works with any MLLM, including models that do not support native grounding.
> Furthermore, our method provides the confidence score for keypoints, which native grounding does not, and these scores substantially strengthen the SAM2 prompting stage.
> Together, these properties lead to significantly higher performance, particularly on the temporally challenging ReasonVOS benchmark (Tab. B).
>
> **Table B: Comparison of training-free video grounding methods using Qwen2.5VL-7B + SAM2.**
> | Method | Ref-DAVIS | ReasonVOS | ReVOS
> | ------ | ----- | ----- | ----- |
> | Native Grounding | 64.8 | 48.0 | 44.9 |
> | Loc-Head | 64.6 | 41.1 | 47.0 |
> | DecAF | 75.2 | 63.9 | 54.2 |
>
> We measure the latency over Ref-DAVIS benchmark with CUDA-synchronized timing and report mean ± std.
> As shown in the comparison table (Tab. C), DecAF requires 8.5s for the MLLM stage and 25.2s for the SAM2 stage, whereas VRS-HQ takes 0.7s and 3.3s, respectively.
> This gap is expected, given the fundamentally different goals of the two methods.
> DecAF processes high-resolution image tiles to preserve spatial cues for training-free grounding, while VRS-HQ employs drastically reduced visual tokens through aggressive token merging.
> The main bottleneck in our full pipeline is SAM2 (Tab. D), as our current implementation prompts SAM2 in a per-object loop rather than a batch-optimized form. This is an engineering limitation rather than a conceptual one and can be substantially optimized, but such system-level optimization lies out of the scope of this work.
>
> **Table C: Overall pipeline latency comparison with a training-based method.**
> | Component | Latency (s) |
> |----------|--------------|
> | (DecAF) MLLM Pipeline| 8.502 ± 0.114 |
> | (DecAF) Mask Pipeline | 25.242 ± 9.296 |
> | (DecAF) All Pipeline | 33.744 ± 9.294 |
> | (VRS-HQ) MLLM Pipeline | 0.713 ± 0.343 |
> | (VRS-HQ) Mask Pipeline | 3.260 ± 0.921 |
> | (VRS-HQ) All Pipeline | 3.972 ± 1.065 |
>
> **Table D: Detailed latency breakdown of DecAF.**
> | Component | Latency (s) |
> |----------|--------------|
> | Object-focused MLLM (Video) | 0.507 ± 0.017 |
> | Object-focused MLLM (Frame) | 3.696 ± 0.250 |
> | Background-focused MLLM (Video) | 0.509 ± 0.011 |
> | Background-focused MLLM (Frame) | 3.469 ± 0.050 |
> | Object Choice MLLM | 0.321 ± 0.250 |
> | Attention Fusion | 0.002 ± 0.005 |
> | Point Sampling | 0.002 ± 0.001 |
> | SAM2 | 25.238 ± 9.296 |

---

### Author Response · Authors · 2025-12-04
**Summary of Rebuttal (1/2)**

**Summary of this work.**

This is the first comprehensive work to explore training-free adaptation of MLLMs for video segmentation through our novel attention-fusion method.
While our attention-fusion method enables MLLMs to directly provide coarse video segmentation masks, our attention-guided SAM2 prompting method further allows the generation of dense masks.
Our method achieves performance comparable to training-based methods and even surpasses them on ReasonVOS, a benchmark requiring complex reasoning and evaluated in a zero-shot setting.

**Summary of discussion.**

Initially, the paper received mixed reviews. `eWqc` (Score 8) and `M2r8` (Score 6) provided positive reviews, whereas `SdCm` (Score 2) and `h74W` (Score 4) provided negative ones.
In response to the reviewers' comments, we present additional experimental results and visualizations to further improve the completeness of this work.
During the discussion period, the concerns raised by `SdCm`, the reviewer who initially gave a score of 2, were thoroughly addressed, and the reviewer stated that they "plan to raise my score to 6".
Specifically, reviewer `SdCm` noted the following:
"showing greater potential and robustness in the video domain due to the proposed fusion mechanisms."
Below, we summarize the major and minor concerns raised by the other reviewers and how we addressed them.

**Summary of the concerns and responses.**

Here, we present a summary of the reviewers' concerns and our responses.
For ease of reference, we include the original response numbers so that each concern can be cross-checked with the detailed responses.

**Major comments**:
* Limited novelty (1) (`SdCm` W4-1).
Reviewer `SdCm` stated that our attention fusion method is a simple operation.
Although the design of our method is simple and intuitive, the proposed decomposed attention fusion (DecAF) delivers non-trivial and meaningful results. Compared to the existing training-free visual grounding method Loc-Head, our method has a clear advantage: it generalizes reliably across different MLLMs and across both image and video domains, as it does not rely on data-specific or model-specific heuristics. Our method outperforms Loc-Head not only on image benchmarks but also across five video benchmarks. In particular, when temporal reasoning is required, our method significantly outperforms Loc-Head (63.9 vs. 41.1 on ReasonVOS). During the discussion period, the reviewer noted that "they demonstrate that the proposed method achieves competitive performance compared to LocHead on image tasks, while showing greater potential and robustness in the video domain due to the proposed fusion mechanisms."
* Limited novelty (2) (`h74W` W1).
The reviewer `h74W` tackles the straightforward modification in attention rollout mechanism. However, our contribution lies in a new formulation that reframes video segmentation as a video QA task, enabling MLLMs’ internal attention flow to serve as a direct grounding signal. Together with vision-aware normalization for attention rollout, decomposed attention fusion, and SAM2-based attention-guided prompting, we build the first complete and reliable training-free pipeline for video segmentation. These components are essential and provide a strong foundation for future research on training-free adaptation of MLLMs.


* Missing generalization comparison (`h74W` W4). The reviewer pointed out that we miss the demonstration of generalization ability to scenarios with no training data. We presented the results on ReasonVOS, which is a zero-shot evaluation dataset, and our method outperforms the existing training-based methods in a large gap, even the method employing the same MLLM with sophisticated RL-based training strategy. Therefore, the provided experimental results strongly support the generalization capability of our method.

* Missing baselines (`SdCm` W1). Some recent MLLMs offer native grounding capabilities; for example, Qwen2.5VL can directly produce grounding results (boxes/points), but we initially missed its results. Since it demonstrated only in the image domain, we experimented with several adaptations for the video domain and constructed the fairest and best-performing baseline for comparison. Our method still significantly outperforms these native grounding results, demonstrating its effectiveness in the underexplored video domain.

* Prompt robustness (`M2r8` W2, `h74W` W3). The reviewers are concerned that the paper lacks an ablation study evaluating the method under diverse prompts. We experimented with three prompt templates for each of the object-focused, background-focused, and object category choice prompts. The consistent results demonstrate the robustness of our method and show that it is not overfit to any specific prompt.

---

> ### Author Response · Authors · 2025-12-04
> **Summary of Rebuttal (2/2)**
>
> **Minor comments**:
>
> * Missing analysis of inference time (`eWqc` W2, `SdCm` W2, `M2r8` W1, `h74W` W2). All reviewers noted that inference time analysis was missing. We provided a thorough latency analysis, including a comparison with a training-based method and a component-wise breakdown.
> Our focus lies on achieving stable and direct visual grounding from any MLLM, independent of its native grounding support; thus, inference speed is not the primary objective.
> Furthermore, it is the first training-free method to achieve comparable performance and even surpass training-based ones on a zero-shot evaluation dataset.
> During discussion, we provided detailed analysis and identified the main bottlenecks, outlining clear directions for improving efficiency in future work.
>
> * Loc-Head results with LLaVA-OV-7B (`eWqc` W1). Initially, we provided Loc-Head results using LLaVA-7B. Following the reviewer's request, we additionally provided the results using LLaVA-OV-7B, and our method still outperforms Loc-Head.
>
> * Results on image segmentation benchmarks (`SdCm` W4). A component of our method - contrastive object-background attention fusion - can also be applied in the image domain, and the reviewer requested a comparison with Loc-Head on image benchmarks. We provided experimental results showing that our contrastive attention fusion method outperforms Loc-Head on RefCOCO+, RefCOCOg, and the challenging ReasonSeg. These results further demonstrate the robustness of our method in the image domain.
>
> * Possibility to handle challenging cases. Reviewers asked whether our method can handle several challenging cases: (1) selecting a single object among multiple similar objects (`SdCm` W5-1), (2) segmenting a specific part of an object, such as a person's shirt (`SdCm` W5-2, `M2r8` W4), and (3) handling non-explicit object expressions where the target is not explicitly stated in the expression (`M2r8` W3). We provided both quantitative and qualitative results demonstrating that our method effectively handles all these cases.

---

### Meta-Review · Area_Chair_UaiV · 2026-01-04

**Summary:**

This paper proposes DecAF for the training-free video referring segmentation task. The proposed method mainly consists of two stages: contrastive object–background attention rollout, and sparse video and dense frame fusion. In addition, SAM2 is employed for further segmentation refinement. The method achieves performance comparable to training-based approaches on multiple video object segmentation benchmarks. Reviewers highly appreciated the novelty and inspiration of the proposed method, affirmed the experimental results, and gave high evaluation scores.

**Reviewer Concerns:**

The reviewers mainly raised concerns regarding the novelty, generalization and robustness, as well as the lack of baseline comparisons in the paper. Through the authors’ responses, all these questions were addressed, and reviewer SdCm expressed approval of the authors’ replies.

**Reviewer Scores:**

Reviewer SdCm will increase their score, while the other reviewers tend to maintain their original scores.

---

### Decision · Program_Chairs · 2026-01-26

Accept (Poster)